# Everybody Needs a Little HELP: Explaining Graphs via Hierarchical Concepts

## Abstract

Graph neural networks (GNNs) have led to major breakthroughs in a variety of domains such as drug discovery, social network analysis, and travel time estimation. However, they lack interpretability which hinders human trust and thereby deployment to settings with high-stakes decisions. A line of interpretable methods approach this by discovering a small set of relevant *concepts* as subgraphs in the last GNN layer that together explain the prediction. This can yield oversimplified explanations, failing to explain the interaction between GNN layers. To address this oversight, we provide HELP (**H**ierarchical **E**xplainable **L**atent **P**ooling), a novel, inherently interpretable graph pooling approach that reveals how concepts from different GNN layers compose to new ones in later steps. HELP is more than 1-WL expressive and is the first non-spectral, end-to-end-learnable, hierarchical graph pooling method that can learn to pool a variable number of arbitrary connected components. We empirically demonstrate that it performs on-par with standard GCNs and popular pooling methods in terms of accuracy while yielding explanations that are aligned with expert knowledge in the domains of chemistry and social networks. In addition to a qualitative analysis, we employ concept completeness scores as well as concept *conformity*, a novel metric to measure the noise in discovered concepts, quantitatively verifying that the discovered concepts are significantly easier to fully understand than those from previous work. Our work represents a first step towards an understanding of graph neural networks that goes beyond a set of concepts from the final layer and instead explains the complex interplay of concepts on different levels.

## 1 Introduction

Graph neural networks (GNNs) have recently enjoyed increasing popularity and have been successfully applied in a variety of domains, ranging from improved travel-time estimations (Derrow-Pinion et al., 2021) to the multi-billion dollar industry of *de novo* drug design (Xiong et al., 2021). However, their application in safety-critical domains, such as healthcare, remains limited. Their lack of interpretability hinders human trust and thus limits their uptake in practice. Consequently, a variety of *post-hoc* explainability methods have been proposed to allow insights into how and why GNNs produce certain predictions (Ying et al., 2019b; Vu & Thai, 2020; Baldassarre & Azizpour, 2019). Crucially, these methods are limited to finding simplified explanations of a complex model, without explicitly incentivizing the model to produce easily understandable predictions. A recent line of research (Magister et al., 2022; Georgiev et al., 2022) therefore attempts to develop GNN-based approaches that are *interpretable-by-design*. In particular, these methods are *concept-based*—they explain their predictions in terms of high-level *concepts* (e.g., recognizable forms such as an "eye" or "mouth" in an image, or functional groups in a molecular graph) rather than raw input features (pixels for images, single nodes with their full embeddings for graphs).

However, while relevant, these methods—post-hoc or not—suffer from a known problem in interpretability: in order to be understood by humans, the explanations tend to be overly simplistic. In this work, rather than just discovering a set of concepts, we take a step towards understanding how concepts from earlier layers compose to new ones later on. Take, for instance, the graph representation of a molecule (Figure 1a). To predict certain properties, knowing all atoms might not be relevant. Instead, the atoms make up relevant subgraphs—so-called *functional groups*—that are sufficient for the final prediction. A similar property holds for social networks. These often consist of

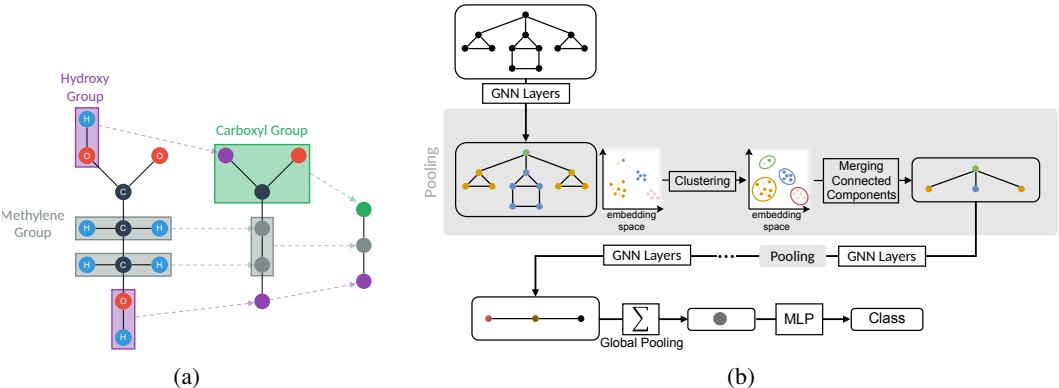

(a)                                                                                        (b)

Figure 1: **(a) Example hierarchy in a molecule.** Meaningful groups of atoms are pooled into a single node representing this *functional group*. **(b) Overview of HELP**. In each pooling step, we apply a number of GNN layers and cluster the resulting node embeddings. Connected components of nodes that were mapped to the same cluster are merged where the new node's embedding is given as the average over the embeddings of all merged nodes. Colors represent node embeddings. Notably, the clustering is always performed over the embeddings of multiple graphs (see Section 3.1).

a set of highly connected *communities*. In many practical tasks, that rely on analyzing large graphs with many outliers, it is often sufficient to reason about the type of these communities or functional groups and the connections between them, rather than reason about all nodes.

In this paper, we provide HELP, a **H**ierarchical **E**xplainable **L**atent **P**ooling method. HELP allows for analysis at different levels of the hierarchical structure of graphs, by repeatedly pooling the input graph to a coarser representation (Figure 1a). More specifically, at each step it executes multiple GNN layers and then merges all connected components belonging to the same cluster in the space of node embeddings. This approach is *interpretable-by-design*. By analyzing which nodes are pooled together, we can identify the relevant substructures for the model's decision. When applied to domains that are less studied, this could not only help to understand the behavior of the model, but by doing so it can lead to new insights about the task (and domain) itself—just like chess players learn new moves from AlphaGo (Willingham).

The contributions of this work are twofold:

1. **HELP**: a novel hierarchical pooling procedure. This is the first non-spectral method that can learn to partition graphs into a variable number of arbitrary connected components end-to-end based on graph structure and features. In addition to being interpretable by design, our technique preserves sparsity (i.e. does not yield fully connected graphs after pooling), increases the receptive field beyond the number of GNN layers and is more expressive than message-passing GNNs.

2. **Concept conformity**: a novel metric to measure the level of noise in a discovered concept.

Using this new conformity metric along with *concept completeness* and comparing discovered concepts against expert domain knowledge in a qualitative evaluation, we demonstrate that the explanations discovered by HELP yield deeper insights while being easier to understand than those from previous work.

## 2    GRAPH NEURAL NETWORKS

GNNs are typically formulated in terms of *message passing* (Battaglia et al., 2018). The aim is to update the embedding of each node based on the set of *messages* which are computed between the node and each of its neighbors. Formally, take a graph $\mathcal{G} = (V, E, X)$, where $E \subseteq V^2$ is the set of edges and without loss of generality we assume the nodes $V = [|V|]$ are identified by the natural numbers 1 to $|V|$. $X := (\boldsymbol{x}_1^0, \ldots, \boldsymbol{x}_{|V|}^0)$ denote node-level input features $\boldsymbol{x}_v^0 \in \mathbb{R}^{d_0}$. Each layer $\ell+1$

then computes new node features

$$\mathbf{x}_v^{\ell+1} = \phi_{\ell+1}\left(\mathbf{x}_v^\ell, \bigoplus_{u \in N(v)} \psi_{\ell+1}\left(\mathbf{x}_u^\ell, \mathbf{x}_v^\ell\right)\right), \tag{1}$$

where $\phi_\ell, \psi_\ell$ represent learnable functions, $N(v)$ denotes the *neighbors* of node $v$ and $\oplus$ is some permutation-invariant function like the sum. In this work, we focus on making one prediction for the entire graph. This can be achieved by first applying the permutation invariant function GLOBALPOOL, to pool the embeddings of all nodes in the last layer $L$, and then computing the final output $f(\mathcal{G})$ as a learnable function $g$ of $f(\mathcal{G}) = g\left(\text{GLOBALPOOL}(\text{GNN}(\mathcal{G}))\right)$, where $\text{GNN}(\mathcal{G}) := \left(\boldsymbol{x}_1^L, \ldots, \boldsymbol{x}_{|V|}^L\right)$.

## 3 Hierarchical Explainable Latent Pooling

The underlying idea of our approach is applying a series of *pool blocks* to the input graph, progressively creating coarser versions (Algorithm 1 and Figure 1b). Each pool block first applies multiple GNN layers. We then cluster the generated node embeddings, merging all connected components of nodes with the same cluster assignment. Only merging nodes in a connected component allows us to maintain the high-level graph structure. For example, if we have the same functional group at different positions in a molecule, we want to merge each occurrence into one node but not both into the same.

### 3.1 Clustering

For the clustering, we choose k-means using Lloyds algorithm for two reasons. Firstly, it is comparatively cheap to compute, which is crucial as it needs to be calculated many times during training. Secondly, it proved itself to give the best concepts when clustering node embeddings in prior work (Magister et al., 2021).

Using k-means assumes we know the exact number of concepts in advance. While this is a hyperparameter we need to tune, in practice, our method is not overly sensitive to it as long as we slightly overestimate. This is because required concepts can often be split up into more fine-grained ones, even if their distinction is not relevant for the final prediction. For instance, a house motif with one or two neighbors could be mapped to different concepts even though the same might be sufficient for the final classification. Similarly, in computer vision, this would be like mapping blue and green eyes to different concepts even though that information is not necessary to detect a face. Nevertheless, it is important to emphasize that even with a fixed number of clusters, we can still learn variable size graphs based on structure and node features. On the one hand, the number of pooled nodes can be lower than the number of concepts as clustering is performed over multiple graphs and some of them might only have nodes in some of the clusters. On the other hand, a pooled graph can contain more nodes than the number of concepts because the same concept can be mapped to multiple new nodes if they are disconnected in the graph.

In Algorithm 1, we directly apply k-means clustering in the pooling step of each batch. However, this poses one important challenge: certain concepts might not be present in a particular batch. In this case, assuming the same number of clusters would lead to splitting up clusters that were combined in other batches. However, whereas the overall number of concepts may vary as described in the paragraph above, it is crucial that their meaning remains the same between batches. Otherwise, some motifs could sometimes be split up into multiple nodes and sometimes be merged into one. This creates a moving target making learning significantly harder for the subsequent GNN layers.

**Global Clustering**  One way to alleviate this issue would be storing the embeddings of each batch over the whole epoch, then calculating the clustering based on all embeddings and keeping the centroids fixed until the next epoch starts and the process can be repeated. However, this creates a moving target where the outcome of all batches depends on the batches of the last epoch. When the epoch ends, there is a sudden jump in behavior.

**Merging Clusters**  We therefore opt for merging centroids below a certain distance threshold. The underlying idea is that if some concepts are missing, existing clusters will be split up as described

above. However, we would still expect the resulting centroids in the same cluster to be relatively close to each other and therefore hope we could merge them with the right threshold, eliminating the impact of the missing concept. An important remark is that the scale of the embeddings varies significantly during training. Thus, we make our approach approximately *scale-invariant* by giving the distance threshold as a percentage of the distance between the two farthest centroids. In Algorithm 1, this could be incorporated by modifying the KMEANS procedure accordingly.

---

**Algorithm 1** Inference for one batch using HELP. The gray area represents a single pooling step as visualized in Figure 1b. Note that for notational convenience we simply assume the node ids to be unique among graphs (e.g. $V_1 \ni 1 \neq 1 \in V_2$) rather than defining the node ids as tuples $(j, k) \in V_j$. CONCOMP: $E \rightarrow (V \rightarrow \mathbb{N})$ denotes a function that maps a set of edges to a function which maps each node to a unique id of the connected component it belongs to. KMEANS: takes a number of clusters $k$ along with a set of nodes and their embeddings and returns a function mapping each node to a cluster id (with nodes from different graphs being distinct as mentioned above).

---

**Require:** GNNs $\text{GNN}_s, s \in [n_{\text{blocks}} + 1]$ ▷ A GNN for each of $n_{\text{blocks}}$ pool block (plus a final one)
**Require:** numbers of clusters $k_s, s \in [n_{\text{blocks}}]$
**Require:** MLP $g$
**Require:** data batch $\{(V_i, E_i, X_i) \mid i \in [b]\}$ ▷ Where $b$ denote the batch size
 **for** $s = 1, \ldots, n_{\text{blocks}}$ **do** ▷ Iterate over all pool blocks
  **for** $i \in [b]$ **do**
   $X_i \leftarrow \text{GNN}_s(V_i, E_i, X_i)$
  **end for**
  $c \leftarrow \text{KMEANS}(k_s, \{(V_i, X_i)_{i \in [b]}\})$ ▷ cluster ids $c : \left(\bigcup_{i \in [b]} V_i\right) \rightarrow [k_s]$
  **for** $i \in [b]$ **do**
   $q \leftarrow \text{CONCOMP}(\{(u, v) \in E_i \mid c(u) = c(v)\})$ ▷ connected comp. ids $q : V_i \rightarrow \mathbb{N}$
   $V_i \leftarrow [\max_{v \in V_i}\{q(v)\}]$
   $X_i \leftarrow \left(\frac{1}{|q^{-1}(1)|} \sum_{v \in q^{-1}(1)} \boldsymbol{x}_v, \ldots, \frac{1}{|q^{-1}(|V_i|)|} \sum_{v \in q^{-1}(|V_i|)} \boldsymbol{x}_v\right)$
   $E_i \leftarrow \{(q(u), q(v)) \mid (u, v) \in E_i\}$
  **end for**
 **end for**
 **for** $i \in [b]$ **do**
  $X_i \leftarrow \text{GNN}_{n_{\text{blocks}}+1}(V_i, E_i, X_i)$
 **end for**
 **return** $(g(\text{GLOBALPOOL}(X_1)), \ldots, g(\text{GLOBALPOOL}(X_b)))$

---

## 3.2 GRADIENT FLOW

The method described so far has clearly defined gradients from the inputs to the final predictions, which is required for backpropagation. This is, because the new node embeddings are always a scaled sum of the previous node embeddings that were pooled together. However, one may argue that these gradients do not describe change in the "direction" of the clustering itself, i.e., how the loss would change if an additional node was considered part of the same cluster or one of the current nodes was no longer considered part of it. In Section 5.3, we therefore also evaluate using a Monte Carlo estimate of a smoothed loss function instead. In particular, we add small random perturbations to all node embeddings in each sample. Then, we find a hyperplane in the loss function that goes through all of these points and use its gradient for updating the weights. This is detailed in Appendix D. Smoothing the discontinuities that exist wherever changing a node embedding would lead to a different clustering allows us to explicitly optimize the clustering rather than just the embeddings for the current clustering.

## 4 EXPERIMENTAL DESIGN

### 4.1 DATASETS

**Synthetic Datasets** As commonly used synthetic datasets for explainability like BA-Shapes and BA-Community (Ying et al., 2019b) are not designed for graph-level predictions or to contain intu-

itive hierarchies, we propose a *Synthetic Hierarchical* dataset. Graphs are constructed from multiple *low-level motifs* (e.g., a triangle) that are each interpreted as a single node and separated by *intermediate nodes* of a different color to make up a *high-level motif* (see Figure 4). The goal is to predict the high-level motif along with the set of low-level motifs. Additionally, we propose a *Synthetic Expressivity* dataset where the goal is to predict if the graph consists of one or two circles. This cannot be determined by 1-WL expressive methods like message-passing GNNs (Xu et al., 2019).

**Common Benchmarks**  In addition to the synthetic datasets above, we evaluate our approach on three real-world datasets. In REDDIT-BINARY (Yanardag & Vishwanathan, 2015), each input graph corresponds to a thread on the social network Reddit. Nodes do not have features and represent users whereas an edge between two users implies that one of them replied to the other. The goal is to classify whether a thread is question-answer-based or discussion-based. For the other two, Mutagenicity (Kazius et al., 2005) and BBBP (Martins et al., 2012), the inputs are graph representations of molecules. The prediction target is a binary classification per graph in both datasets—namely, whether the molecule is mutagenic in the case of Mutagenicity and whether it can penetrate the blood brain barrier in the case of BBBP.

## 4.2 EVALUATION METRICS

**Concept Completeness**  Yeh et al. (2020) aims to determine the expressiveness of the discovered concepts for the given goal. It does so by training a model to predict the target only from the discrete set of concepts. The accuracy this model can reach is called *completeness*. Magister et al. apply this to graphs by training a decision tree to predict a node's classification from the I.D. of the concept it was mapped to. For the graph classification case, they still aim to predict the graph's class from a single node and take the average accuracy over all nodes of the graph. As the combination of concepts in a graph can be highly relevant, we propose to predict it's class from the multiset of all concepts rather than predicting it for each concept separately. This also yields decision trees explaining how concepts combine to a final prediction. To account for the hierarchies discovered by our method, we compute the completeness after each pooling step.

**Concept Conformity**  The second desired property of concepts is that they are easy to understand with as few as possible outliers. Magister et al. (2021; 2022) measure this in terms of *purity*. Each concept is represented by the most frequent $k$-hop neighborhood of nodes mapped to it and the purity is given by the average graph edit distance between this representation and the $k$-hop neighborhood of a node mapped to the concept.

As an alternative, we propose to measure *concept conformity*. Intuitively, we would like each concept to contain only a small set of subgraphs, all of which appear frequently enough to be relevant for the concept. Other subgraphs, that only occur rarely could be considered noise that makes the concept impure (see Figure 2 for an example). We therefore define the conformity of a concept in a given pooling layer as the percentage of subgraphs that make up at least a fraction $t$ of the overall subgraphs. An empty concept has the perfect conformity of 100%. Formally, this is given by

$$\text{conf}(c) = \frac{1}{o_c} \sum_{j=1}^{n_{\text{sub}}} o_c^{(j)} \mathbb{1}_{[to_c, \infty)}\big(o_c^{(j)}\big) \qquad (2)$$

where $n_{\text{sub}}$ denotes the number of pairwise non-isomorphic subgraphs that got pooled together, $o_c^{(j)}$ the number of times that a subgraph isomorphic to $j \in [n_{\text{sub}}]$ was mapped to concept $c \in [n_{\text{clust}}]$ and $o_c := \sum_{j=1}^{n_{\text{sub}}} o_c^{(j)}$ the the total number of subgraphs that were mapped to cluster $c$. Note that we leave out the dependence of all variables on the pooling step to avoid unnecessarily convoluted notation. The conformity of a layer is then given as the average conformity over all non-empty ($o_c \neq 0$) concepts in that layer. An obvious drawback of this definition is that it requires choosing the threshold $t$ which we define as $t := 10\%$ in our case.

This definition has three major advantages over concept purity. Firstly, as HELP can learn the exact part that belongs to a concept[1], we simply count the number of nonisomorphic subgraphs mapped to the same concept and no longer rely on the graph edit distance. Note that even a graph edit distance

---
[1]In Appendix F.1, we compare this to using the $k$-hop neighborhood.

of one could easily make the difference between two functional groups with vastly different impact while other concepts might have many unimportant nodes in the $L$-hop neighborhood despite all of them representing the same concept. Additionally, we take into account discrete node features (like atom type), as, for example, a chain of three carbon atoms should not be considered the same as an $NO_2$ group. For all pooling steps after the first one, we use the concept I.D. instead. Finally, concept purity assumes that only a single subgraph should be mapped to each concept. However, in practice there might be different substructures that have the same meaning for the final prediction. For instance, for many molecular prediction tasks the length of a chain of carbon atoms might be close to irrelevant for the predicted property. Consequently, mapping these subgraphs to the same concept would be desirable as it gives us insights into the dynamics of the domain while decreasing the number of concepts we need to interpret.

## 5 RESULTS & DISCUSSION

We begin by demonstrating that HELP performs as well as DiffPool (Ying et al., 2019a), ASAP (Ranjan et al., 2020) and a standard GCN (Kipf & Welling, 2017) in terms of accuracy while being more than 1-WL expressive. At the same time, it outperforms GCExplainer (Magister et al., 2021) and a definition of concepts in DiffPool in terms of concept conformity and completeness. This quantitative evaluation is complemented by a qualitative assessment in which we first analyze some general observations regarding the composition of concept hierarchies in the example of our Synthetic Hierarchical dataset. We then identify concepts discovered in BBBP, Mutagenicity and REDDIT-BINARY which align with expert domain knowledge. Finally, we conduct an ablation to show that our version of HELP performs on-par with more computationally expensive variations using global clustering or hyperplane gradient approximation.

### 5.1 QUANTITATIVE ANALYSIS

We start by comparing HELP to the simple GNN baseline GCN (Kipf & Welling, 2017) along with two popular pooling approaches: DiffPool (Ying et al., 2019a) and ASAP (Ranjan et al., 2020). First, we note that HELP outperforms DiffPool and a standard GCN in our Synthetic Expressivity dataset, indicating that it indeed is more than 1-WL expressive as detailed in Appendix A.1. For the rest of this section, we will focus on the other datasets.

We observe that when using the same general GNN architectures for all methods[2], the accuracy of HELP almost always lies within one standard deviation of the best approach. While this implies on-par performance with previous methods, the primary contribution of HELP is its explainability. Since our technique is the first to extract hierarchical concepts, we compare it to the overall concepts discovered by the *post-hoc* method GCExplainer on our GCN baseline. Additionally, we extract concepts from the hierarchical pooling method DiffPool by defining the concept of each node as the new node it got pooled to with the highest weight. HELP significantly outperforms both of those baselines in terms of concept conformity. The only exception is our Synthetic Expressivity dataset. However, note that neither of them is able to solve this benchmark, rendering the discovered concepts meaningless. Additionally, while the conformity of 0 and a completeness close to the probability of the most likely class may seem like a negative result at the first glance, note that this synthetic dataset is purposefully designed in a way that any connected component of nodes has the same meaning. Therefore, mapping all nodes to the same concept is the ideal strategy (as detailed in Appendix A.1), despite resulting in these scores.

Whereas DiffPool achieves significantly lower completeness, the slightly higher scores of GCExplainer can be attributed to the fact that the concepts are calculated for the last GCN layer whereas for the hierarchical methods, the reported concepts are only those after the first pool block. These will be combined to more meaningful concepts in the subsequent layers. On the other hand, the conformity scores of GCExplainer are significantly lower. Figure 2 reveals how profound the impact of this is for explainability. While the concepts discovered by GCExplainer are slightly more predictive of the final outcome, they are significantly harder to understand.

**Feature completeness** Whereas previous work in explainable GNNs focused only on the last layer, our hierarchical analysis after each pooling block naturally extends to the question: What is the

---

[2]See Appendix B.1 for details

Table 1: **Test accuracy, completeness and conformity scores in comparison to other methods.** All values are given as mean (in percent) and standard deviation over three models with different seeds. As conformity and completeness should only be viewed together, we underline where there average is highest. We account for the stochasticity of decision trees by calculating completeness scores three times per model. For brevity we only report conformity and completeness scores after the first pooling layer for HELP and DiffPool, all values can be found in Tables 5 and 6.

| | **HELP** (Ours) | | | **DiffPool** | | | **ASAP** | | | **GCN** | ( + GCExpl.) | | **Feature** |
| | Acc. | Comp. | Conf. | Acc. | Comp. | Conf. | Acc. | Comp. | Conf. | Acc. | Comp. | Conf. | **Comp.** |
|---|---|---|---|---|---|---|---|---|---|---|---|---|---|
| **Synth. Hier.** | $99.9\pm0.2$ | $\mathbf{100.0\pm0.0}$ | $\mathbf{99.8\pm0.4}$ | $\mathbf{100.0\pm0.0}$ | $27.0\pm0.0$ | $0.0\pm0.0$ | $96.9\pm4.8$ | n/a | n/a | $\mathbf{100.0\pm0.0}$ | $\mathbf{100.0\pm0.0}$ | $16.8\pm1.8$ | $46.9\pm0.0$ |
| **Synth. Exp.** | $\mathbf{100.0\pm0.0}$ | $52.3\pm0.0$ | $0.0\pm0.0$ | $53.5\pm0.0$ | $\mathbf{53.5\pm0.0}$ | $0.0\pm0.0$ | $93.9\pm0.2$ | n/a | n/a | $53.5\pm0.0$ | $\mathbf{53.5\pm0.0}$ | $74.4\pm0.0$ | $53.5\pm0.0$ |
| **Mutag.** | $77.0\pm2.3$ | $73.7\pm2.7$ | $\mathbf{83.6\pm0.3}$ | $78.7\pm0.6$ | $53.6\pm0.0$ | $42.9\pm0.0$ | $76.2\pm1.7$ | n/a | n/a | $80.5\pm0.7$ | $\underline{77.5\pm2.4}$ | $16.5\pm10.1$ | $62.1\pm0.7$ |
| **BBBP** | $85.0\pm1.6$ | $80.8\pm1.4$ | $\mathbf{84.8\pm1.4}$ | $82.0\pm5.6$ | $77.1\pm0.4$ | $0.0\pm0.0$ | $85.2\pm1.5$ | n/a | n/a | $84.9\pm3.1$ | $86.0\pm1.6$ | $5.8\pm6.0$ | $79.4\pm0.9$ |
| **REDDIT-BIN**[3] | $88.7\pm2.2$ | infeas. | $\mathbf{96.2\pm0.4}$ | $\mathbf{93.9\pm0.7}$ | infeas. | $93.0\pm2.6$ | infeas. | n/a | n/a | $89.1\pm0.9$ | infeas. | infeas. | infeas. |

completeness of the *input* concepts (i.e., the set of node features without any graph information)? We report these *feature completeness* scores in Table 1 and argue that they should be reported as an important baseline when using concept completeness scores on graphs. For datasets like BBBP where the node features alone already determine the outcome with high accuracy, methods like GCExplainer, which give one concept per node, might be of limited benefit. Note that HELP yields less concepts than nodes which means that the individual concepts are more meaningful than the raw input vectors, even with comparable completeness scores.

## 5.2 QUALITATIVE ANALYSIS

**General observations** Since human understanding is a highly subjective matter, metrics can only yield limited insights into the usefulness of discovered concepts and a qualitative analysis is crucial. We start with some general observations using examples from our Synthetic Hierarchical dataset depicted in Figure 2. Firstly, we note that the same subgraph can be mapped to different concepts if the impact on the outcome is different. For instance, bodies of low-level pentagons are mapped to different concepts depending on their degree which impacts the class. Secondly, while many concepts only have one meaning, some are better understood when interpreting them as representing *either* the first subgraph *or* the second where both have the same impact on the prediction or at least determine it in combination with the other concepts that can occur with it. For example, in our Synthetic Hierarchical dataset the completeness score of 100% implies that this must hold despite "center of house" and "body of deg 3 pentagon" intuitively looking unrelated. Finally, concepts in later pooling steps can be more specific (e.g., splitting up center of house into centers of deg 2 and deg 3 houses), or more general (e.g., combining different concepts into "at least 2 neighboring houses").

**Domain-specific concepts** To demonstrate the general applicability of HELP to more realistic tasks, we plot the pooled graph distribution with some subgraph examples for all datasets in Appendix F.3. For *Mutagenicity* (Figure 13), we find that concept 1 represents the $NO_2$ group and concept 4 represents aromatic rings—both of which are known to have a strong impact on the mutagenicity of a molecule (Kazius et al., 2005). Additionally, we identify concepts for the common hydroxy group (14) and for most likely irrelevant hydrogen atoms (16). Important concepts discovered in *BBBP* (Figure 11) include aromatic rings (3), oxygen atoms (9), nitrogen atoms (2, 4, 7) and carbon atoms close to them (1, 12, 13). All of these strongly influence blood-brain-barrier penetration (Pajouhesh & Lenz, 2005; He et al., 2018). As expected (Ying et al., 2019b), the concepts discovered for *REDDIT-BINARY* (Figure 15) contain both, tree-like structures with few neighbors, indicating a discussion between a number of users (e.g., 1, 5, 14 and 17) and structure with one central user to whom many others reply, indicating a question-answer based thread (e.g., 7, 11, 12 and 18).

## 5.3 ABLATION

**Global Clustering** In the baseline version of HELP which we discussed so far, we perform clustering batch-wise with merged centroids as defined in Section 3.1. As shown in Table 2, performing

---

[3]For computational feasibility we only use 50% of the test set for conformity scores. ASAP could not be run in the given configuration as even small batch sizes did not fit into 80GB of GPU memory.

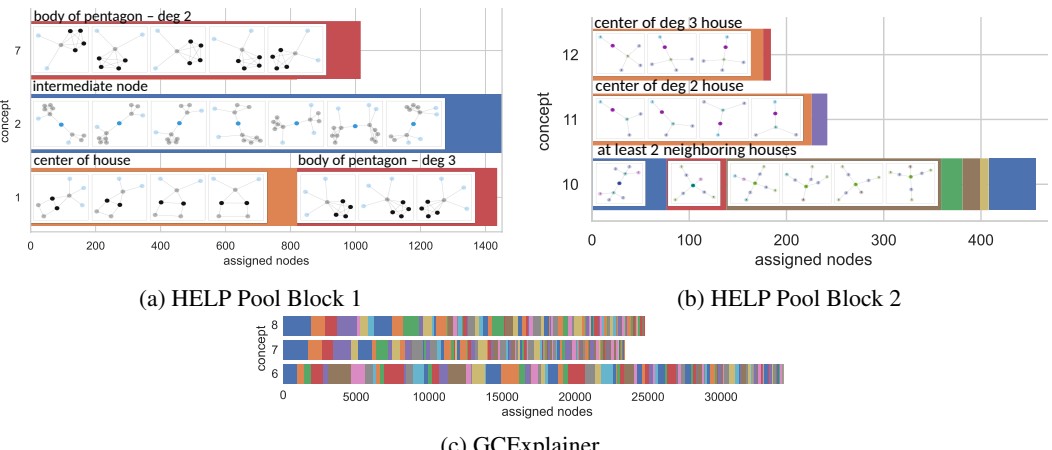

(a) HELP Pool Block 1  (b) HELP Pool Block 2

(c) GCExplainer

Figure 2: **Excerpt of concepts found in our hierarchical dataset and which pooled subgraphs were assigned to them**. Each bar of one color shows a set of isomorphic subgraphs mapped to a concept. The bars contain random example $L$-hop neighborhoods (transparent) of the pooled subgraph (solid). The node color represents the features. By using the number of subgraphs rather the number of nodes (x-axis), these plots illustrate the intuition of concept conformity, where any colored parts that make up less than $10\%$ of the concept's total bar are considered noise. Extended versions are given in Appendix F.3.

Table 2: **Test accuracy and completeness scores of ablations.** Reported as in Table 1.

| | Baseline | | | Global Clustering | | | Hyperplane | | |
|---|---|---|---|---|---|---|---|---|---|
| | Acc. | Comp. | Conf. | Acc. | Comp. | Conf. | Acc. | Comp. | Conf. |
| **Synth. Hier.** | **99.9**±**0.2** | **100.0**±**0.0** | 99.8±0.4 | 99.6±0.7 | **100.0**±**0.0** | **100.0**±**0.0** | 97.9±2.7 | **100.0**±**0.0** | **100.0**±**0.0** |
| **Mutag.** | 77.0±2.3 | 73.7±2.7 | 83.6±0.3 | **78.7**±**2.0** | 74.4±1.4 | **84.2**±**1.7** | 78.0±0.9 | 72.8±1.4 | 83.0±1.2 |
| **BBBP** | 85.0±1.6 | 80.8±1.4 | **84.8**±**1.4** | **86.2**±**2.2** | 82.1±2.7 | 84.4±2.4 | 84.6±3.4 | 81.0±1.3 | 84.0±2.0 |

clustering on the whole batch would perform marginally better. For concept completeness and conformity it is important to note that this variation has a small advantage because it was trained on clusters that were determined by the whole training set, whereas the other methods are also tested on such a clustering, but trained on clusters from only a single batch. However, whereas the differences in the metrics all lie within one standard deviation, global clustering requires keeping all final node embeddings from all pooling steps in memory at once. This proves prohibitive on larger datasets like REDDIT-BINARY.

**Hyperplane Gradient Approximation**  Applying the hyperplane gradient approximation technique detailed in Appendix D.2 performs extremely similar to not using it. This confirms the hypothesis that node embeddings already form clusters of meaningful concepts when training a GNN end-to-end without explicitly optimizing the clusters (Magister et al., 2021). As the runtime per epoch grows linearly in the number of Monte Carlo samples (in our case with 10 Monte Carlo samples by a factor of 9.5±1.0 over the three tested datasets), we choose the simpler and more computationally efficient approach as baseline.

**Limitations**  We note that the computational cost prohibited an extensive hyperparameter search on our hyperplane gradient approximation approach. We therefore cannot exclude the possibility that with different hyperparameters, explicitly learning the clustering would further improve the results. Moreover, our definition of the conformity score relies on graph-isomorphism for which no polynomial time algorithm is known. In practice, we use a table of WL hashes (Shervashidze et al., 2011) (which can be computed in linear time), and only check isomorphism for collisions. Regardless, this metric still becomes infeasible for larger datasets like REDDIT-BINARY. This is still significantly less expensive than calculating the graph edit distance as proposed by Magister

et al. (2021) who simply ignore graphs above a certain size in order to be able to compute their purity scores.

# 6 RELATED WORK

**Explainable GNNs** *Post-hoc* explainability methods on GNNs can be divided into three categories. *Instance-level* approaches explain the prediction of a single input sample. This is often done by adapting existing explainability techniques to the domain of graphs (Baldassarre & Azizpour, 2019; Pope et al., 2019; Schnake et al., 2022) or—taking inspiration from the seminal work GNNExplainer (Ying et al., 2019b)—by finding a subgraph that has the highest relevance for the prediction (Vu & Thai, 2020; Schlichtkrull et al., 2021; Yuan et al., 2021). In contrast, *model-level* methods (Wang & Shen, 2022; Yuan et al., 2020; Luo et al., 2020) aim to understand what characterizes a particular prediction (e.g., a given class) over the whole dataset. In an attempt to find a balance between those two ideas, *concept-based* approaches (Azzolin et al., 2022; Xuanyuan et al., 2023) explain individual predictions but do so in-terms of human-understandable *concepts* rather than raw input features. Most similar to our method is GCExplainer (Magister et al., 2021), which also defines concepts as clusters in the embedding space but—besides being post-hoc—cannot show hierarchies or which part of a node's neighborhood is actually relevant for the concept.

By trying to cast light on the complex prediction process without ever incentivizing the model's decisions to be made in an easily understandable way, these post-hoc approaches tend to generate explanations that are less complete, less accurate and more prone to human error (Rudin, 2019). As a remedy, Magister et al. (2022) force the final node embeddings to be in $[0, 1]$ and define a concept as all nodes that were mapped to the same, booleanized embedding. In contrast to HELP, this approach represents concepts as $k$-hop neighborhoods rather than only the relevant part. It has limited applicability to the graph-level prediction setting where they average over all final node embeddings, making them no longer close to binary values. (Georgiev et al., 2022) propose an alternative, specifically targeting the field of learning to imitate classical algorithms and assuming to know the possible concepts in advance.

**Hierarchical Graph Pooling** While earlier work in this field deterministically pre-computes which nodes to cluster based on the graph-structure (Defferrard et al., 2016; Rhee et al., 2018), or learns a fixed pooling in a separate, unsupervised step before training the actual model (Subramonian, 2021; Dai & Wang, 2021), we focus on end-to-end-learnable graph pooling approaches. This allows us to explicitly learn concepts that are relevant for the task at hand. More specifically, while there are numerous *spectral* pooling methods (Dhillon et al., 2007; Ma et al., 2019; Bacciu & Sotto, 2019)—which are problematic due to the time complexity of the Eigendecomposition they rely on—we focus on *non-spectral* approaches. In their method *DiffPool*, Ying et al. (2019a) learn weights mapping any input graph to a fully connected graph with a fixed number of nodes. Later methods like ASAP (Ranjan et al., 2020) approach these two major limitations by learning to prune a fixed percentage of nodes, resulting in sparse graphs where the number of output nodes is a fixed percentage of the number of input nodes (Gao & Ji, 2019; Cangea et al., 2018; Lee et al., 2019). In contrast to HELP, the number of nodes after pooling still cannot depend on structure or features of the input. Finally, while not a hierarchical method, Kan et al. (2022) propose a graph pooling approach that is also based on clustering.

# 7 CONCLUSIONS

We present HELP, a novel graph learning method that explains its predictions in terms of hierarchical concepts. We empirically demonstrate that it performs on-par with previous message-passing GNNs and hierarchical pooling methods in terms of accuracy while being more than 1-WL expressive and discovering significantly more precise and less noisy concepts than the previous state-of-the-art. The latter is shown by a qualitative analysis as well as a novel metric to evaluate the *conformity* of concepts. In future work, it would be valuable to explore adding readout layers after each pooling step, replacing the final sum pooling with variations of Logic Explained Networks (Ciravegna et al., 2023), extending HELP to settings requiring node-level predictions, or utilizing it for graph compression. More importantly, we believe HELP paves the way for a more thorough analysis of the interplay between concepts discovered on different layers of GNNs.

REPRODUCIBILITY

Whereas the most important hyperparameters are given in Appendix C.1, the exact commands to reproduce all ablations are given in the `README.md` file of our repository (attached as supplementary material). Additionally, the `analysis.ipynb` notebook contains detailed instructions to easily reproduce the concept/subgraph visualizations used in the qualitative analysis.

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

# A ADDITIONAL PROPERTIES OF HELP

## A.1 EXPRESSIVE POWER

An additional advantage of HELP is that, in contrast to GNNs following the message-passing framework (Xu et al., 2019), it can distinguish between graphs that cannot be distinguished by the Weisfeiler-Lehman graph isomorphism test (WL-test). Figure 3 gives an example of two graphs that this test fails to tell apart. However, it is easy to see that the number of connected components in these graphs differs. As our pooling step includes a connected component search, our algorithm should, in theory, be able to determine which of those two graphs was given as the input. In particular, the GNN layers will produce the same embeddings for all nodes due to the mentioned limitation. The pooling will then map the graph on the left to two nodes and the graph on the right to one node—all of them with the exact same embedding which was calculated by mean pooling nodes with the same embedding. For nonzero embeddings, the final global sum pooling will therefore yield different results (the graph on the left will have the result of the graph on the right times two), allowing the final classification layer to make different predictions.

To empirically demonstrate this, we design a simple synthetic dataset based on the example graphs shown in Figure 3. In particular, we generate graphs with even numbers of nodes between 6 and 40 which consist of either one or two circles. The goal is to predict which of those two is the case. As shown in Section 5.1, we indeed achieve perfect accuracy whereas GCN only reaches the accuracy of always guessing the more likely class.

Note that HELP trivially also can distinguish everything the GNN layers it uses could differentiate on their own. This makes it strictly more expressive than standard GNNs that follow the message-passing scheme. Interestingly, even though it does not solve it perfectly, the fact that ASAP achieves higher accuracy than just predicting the most likely class on our Synthetic Expressivity dataset indicates that it is more than 1-WL expressive as well. To the best of our knowledge, this is not mentioned in the original paper.

## A.2 RECEPTIVE FIELD

In a standard GNN following the message-passing scheme, the *receptive field* of a node is its $L$-hop neighborhood, where $L$ is the number of GNN layers. A big receptive field can be necessary to recognize larger structures in the graph. However, increasing the number of GNN layers not only leads to higher computational cost but also has a harmful effect termed *oversmoothing*. This refers to the phenomenon that with a growing number of GNN layers, the embeddings of all nodes will become similar which leads to a loss of expressive power exponential in the number of layers (Oono & Suzuki, 2020).

Our algorithm alleviates this issue by increasing the receptive field beyond the number of GNN layers. Take, for instance, a graph consisting of multiple substructures where the interaction of those substructures is important but they are separated by long chains of intermediate nodes. An example would be inserting more intermediate nodes in our Synthetic Hierarchical dataset. A standard GNN would require enough layers to cover those long chains. In contrast, our method could already pool after only one or two layers. Most nodes on the chains would then be mapped to the same cluster—

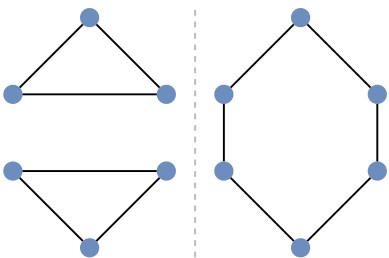

Figure 3: The graph consisting of two triangles (left) and the graph consisting of a hexagon (right) can not be distinguished by standard GNN architectures even though they are not isomorphic.

Table 3: Shared hyperparameters for all datasets

| Parameter | Value |
|---|---|
| Optimizer | Adam (Kingma & Ba, 2015) |
| Learning Rate | 0.001 |
| Weight Decay | 0.0005 |
| Batch Size | 32 for REDDIT-BINARY, 64 otherwise |
| GNN Layers | GCN (Kipf & Welling, 2017) |
| Global Pooling Operation | $\sum$ |
| Activation | LeakyReLU (0.01) |
| Hidden Dimensions | 32 32 [Pool] 32 32 [Pool] 32 4 |
| Train/Test/Validation Split | 80% / 10% / 10% |
| $m$ (no. samples for hyperplane approx.) | 10 |

independent of the length of the chains. Consequently, the nodes on each chain would be mapped to a single node during pooling. After this, a significantly smaller number of layers would be required to cover the whole graph in the receptive field.

# B EXPERIMENTAL SETUP

## B.1 BASELINES

We compare our proposed algorithm to a standard GNN (in our case GCN (Kipf & Welling, 2017)) with the exact same architecture but without the pooling layers. Additionally, we compare it to two popular pooling methods: DiffPool and and ASAP (see Section 6). Whereas none of these approaches was designed with the goal of interpretability, we are still able to calculate concept completeness for DiffPool. Recall that DiffPool has a fixed number of nodes after each pooling step and learns a soft assignment from each input node to these output nodes. We therefore define the concept of an input node as the id of the output node to which the assignment is the strongest. This gives us everything that is required for our concept metrics, such as a multiset of present concepts for completeness. ASAP, on the other hand, only selects some percentage of the most important nodes. There is no straight-forward mapping from these nodes to a global concept. Additionally, note that ASAP and DiffPool both employ more complex GNN layers to achieve state-of-the-art performance and ASAP additionally makes use of a layer-wise *readout*[4]. To enable fair comparison, we instead use the GCN layer for all experimental setups and remove the layer-wise readout.

# C IMPLEMENTATION DETAILS

## C.1 HYPERPARAMETERS

Whereas the most important hyperparameters are given in Tables 3 and 4, the exact commands to reproduce all ablations are given in the `README.md` file of our repository.

## C.2 CLUSTERING

### C.2.1 RESOLVING AMBIGUITIES WHEN MERGING CLUSTERS

Note that it is possible for connected chains of points that should be merged to occur. For example, there could be three points A, B and C where the distance between A and B and between B and C is below the threshold but the distance between A and C is not. We resolve these ambiguities in a permutation-invariant manner by merging the whole connected chain.

---

[4]Essentially, this introduces skip connections from the output of each GNN layer to the final prediction layer.

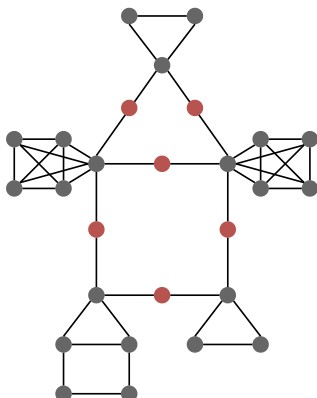

Figure 4: **An example graph from the synthetic hierarchical dataset.** The high-level motif here is a house. It contains two triangles, a house and two fully connected pentagons as lowlevel motifs. The class label is therefore given by (house, {triangle, house, fully connected pentagon}). Intermediate nodes (see Section 4.1) are colored red.

Table 4: Hyperparameters varying between datasets

|  | **Epochs** | $n_{\text{clust}}$ Lvl. 1 | $n_{\text{clust}}$ Lvl. 2 |
|---|---|---|---|
| Synthetic Hierarchical | 10000 | 10 | 15 |
| Mutagenicity | 1000 | 20 | 20 |
| REDDIT-BINARY | 1500 | 30 | 30 |
| BBBP | 5000 | 15 | 15 |
| Synthetic Expressivity | Ours: 100, others: 1000 | 10 | 15 |

## C.3 DATASETS

### C.3.1 HIERARCHICAL DATASET

Note that we insert the intermediate nodes described in Section 4.1 for two reasons. First, they ensure that the combination of low-level and high-level motif can theoretically be deduced. Whereas we view the graph in a certain way based on how it was generated, it is otherwise possible that graphs from different classes are in fact isomorphic. Additionally, the main goal of this dataset is that we already have a good understanding of what concepts to expect when going into the analysis. In particular, we would like to see decoupled concepts for the different low-level motifs which makes the dynamics easier to understand. As our method pools connected components mapped to the same concept, without the intermediate nodes, neighboring high-level nodes would be pooled together if they are mapped to the same concept. Therefore, our algorithm would be forced to learn more complex concepts that depend on the combination of neighboring low-level motifs in order to solve the task. These would be harder to interpret in the analysis.

### C.3.2 MUTAGENICITY AND BBBP

Like previous methods (Magister et al., 2021; 2022), we ignore the edge features in these datasets as they are not supported by the simple GNN layer GCN (Kipf & Welling, 2017). Whereas our method is independent of the message-passing GNN layer and therefore in principle also supports edge features with an appropriate layer, we opt for GCN as a widely used baseline to make our analysis comparable to previous work (Magister et al., 2022; 2021), avoid unexpected side-effects by a complex GNN layer and minimize computational cost. Whereas Mutagenicity has only the atom class as its node labels, BBBP contains various additional properties like *valence* by default. We remove these properties to allow easier visualization and interpretation of the input graphs by someone who is not a domain expert.

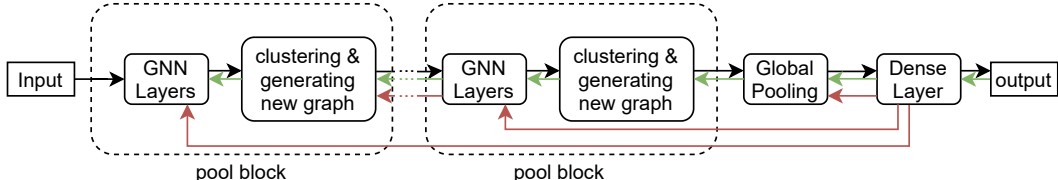

Figure 5: **Gradient flow of our method.** Black arrows denote the forward pass, green arrows show the naïve gradient flow only through the node embeddings (Section D.1) and red arrows denote the gradient flow in our hyperlane-based approximation (Section D.2). In particular, note that the red gradients flow back up to the discontinuous component where the flow is interrupted. Instead, the gradient with respect to the input of these components is approximated from the multiple samples that were propagated from here up to the final dense layer. These approximated input gradients then flow back up to the next discontinuous component, where the process repeats.

## C.4 Visualization

Whilst not required in clean datasets like our Synthetic Hierarchical one, for BBBP and Mutagenicity (plots 13–12), we make visualizations slightly more readable by merging small bar segments into bigger ones. In particular, visualizing examples of these segments reveals that they are generally highly related to bigger segments of the same concept. For example, the same functional group but including an additional carbon atom. While this could still be interpreted by a human, it would require looking at a bigger list of examples and thereby be harder to depict in a single plot. As a remedy, we take all pooled components with at most 500 assigned nodes (starting from the one with the least assigned nodes) and merge them with the next bigger one (if any) that (1) is a proper subgraph (including node features) of the current component and (2) consists of at least 2 nodes (to ensure that relevant structure is preserved). Note that this method could not sensibly be applied to GCExplainer as the different k-hop neighborhoods would likely not be subgraphs of each other.

## D  Differentiability

As mentioned earlier, clustering and merging connected components are inherently discontinuous operations. In this section, we will discuss different remedies that allow us to still learn embeddings that give the desired clusters.

## D.1 Using the Existing Gradient Flow

Before we dive further into different approaches to estimate gradients, it is important to note that they are not strictly necessary to train a model. As the node embeddings of the pooled graphs are always averages over a subset of node embeddings of the previous graph, there exists an uninterrupted gradient flow from the inputs to the final predictions (see Figure 5). However, these gradients are calculated as if the cluster assignments were fixed. In other words, whereas the loss landscape is not zero almost everywhere (as it would be the case for many classical algorithms like sorting, path finding etc. (Pogancic et al., 2020)), it has discontinuities wherever a change in a node embedding would lead to it being assigned to a different cluster. This means that the gradient information does not give us any way of learning which cluster assignment would be better. As learning a good way to coarsen graphs is a central goal of this work, we will discuss different sampling based remedies in the following sections. Nevertheless, since Magister et al. (2021) find that even in standard GNNs, the trained node embeddings form clusters representing meaningful concepts, we still expect this approach to perform reasonably well. Even though we do not explicitly optimize for a good clustering, it will implicitly evolve as the GNN learns to optimally predict the target.

## D.2 Locally Approximating Gradients as a Hyperplane

An alternative approach is motivated by the definition of the gradient as the locally tangent hyperplane as well as Taylor's theorem which tells us that a continuously differentiable function could be

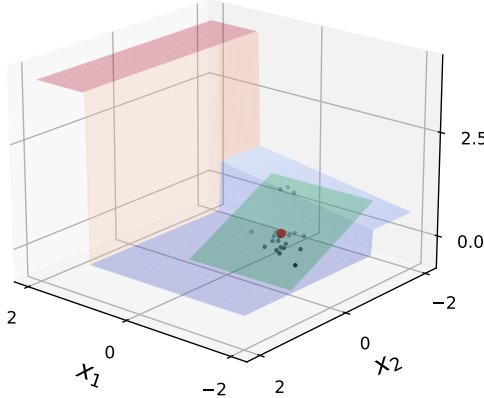

Figure 6: **Visualization of the approximated hyperplane**. For a given point (red) on the function $f$ (blue/red), we take $m$ samples around it (black) and find an approximate hyperplane (green). We then use the gradient of that hyperplane as the approximate gradient at the red point. Note that this example is *overdetermined* as $m > 2$ and $\boldsymbol{x} \in \mathbb{R}^2$. We therefore use the least squares solution to Equation 3 rather then the minimum norm solution. In practice, we have $\boldsymbol{x} \in \mathbb{R}^b$ for some $b \gg m$ and our hyperplane will therefore always go through all sampled points.

locally approximated as a hyperplane. We therefore propose to locally approximate the gradient of some function $f : \mathbb{R}^b \to \mathbb{R}^d$ at point $\boldsymbol{x} \in \mathbb{R}^b$ by evaluating the function on a number $m - 1$ of slightly perturbed points and finding a hyperplane that goes through all of them. A visualization is shown in Figure 6.

Formally, for noise vectors $\boldsymbol{\varepsilon}_1, \ldots, \boldsymbol{\varepsilon}_{m-1} \sim \mathcal{N}(\boldsymbol{0}, \sigma^2 \boldsymbol{I}_b)$ we determine the minimum norm solution $A$ to

$$\begin{bmatrix} \rule{0.5cm}{0.4pt} & \boldsymbol{x}^\top & \rule{0.5cm}{0.4pt} & 1 \\ \rule{0.5cm}{0.4pt} & (\boldsymbol{x} + \boldsymbol{\varepsilon}_1)^\top & \rule{0.5cm}{0.4pt} & 1 \\ & \vdots & & \vdots \\ \rule{0.5cm}{0.4pt} & (\boldsymbol{x} + \boldsymbol{\varepsilon}_{m-1})^\top & \rule{0.5cm}{0.4pt} & 1 \end{bmatrix} \underbrace{\begin{bmatrix} a_{1,1} & \cdots & a_{1,d} \\ \vdots & \ddots & \vdots \\ a_{b+1,1} & \cdots & a_{b+1,d} \end{bmatrix}}_{=:A} = \begin{bmatrix} \rule{0.5cm}{0.4pt} & (f(\boldsymbol{x}))^\top & \rule{0.5cm}{0.4pt} \\ \rule{0.5cm}{0.4pt} & (f(\boldsymbol{x} + \boldsymbol{\varepsilon}_1))^\top & \rule{0.5cm}{0.4pt} \\ & \vdots & \\ \rule{0.5cm}{0.4pt} & (f(\boldsymbol{x} + \boldsymbol{\varepsilon}_{m-1}))^\top & \rule{0.5cm}{0.4pt} \end{bmatrix}$$

(3)

where we append ones to the input to allow for a constant offset in the hyperplane. We can therefore locally approximate $f$ around $\boldsymbol{x}$ as:

$$\hat{f}(\boldsymbol{z}) := A^\top \begin{pmatrix} \boldsymbol{z}_1 \\ \vdots \\ \boldsymbol{z}_b \\ 1 \end{pmatrix}$$

(4)

where we can trivially read off the gradients from $A$.

Whereas in traditional finite difference methods, the distance between the points should be as small as possible to obtain the best possible gradient approximation, this is not the goal in our case. Instead, we need to strike a balance with the chosen variance. On the one hand, the points should be spread out far enough that when close to a discontinuity, some points will likely be on the other side of this "jump". This is what makes our gradient estimations smooth. On the other hand, increasing the variance of the point distribution also increases the variance of our gradient estimates which means that we need more samples to attain stable gradients. Moreover, the estimated gradient should not be overly smooth in order to still be able to find the correct minimum.

Whereas the previously proposed black box differentiation methods (Berthet et al., 2020; Pogancic et al., 2020; Blondel et al., 2020; Dalle et al., 2022) focus on linear programs, our approach has

an intuitive motivation for arbitrary functions. This is particularly useful as in our setting we need to map a mixture of continuous (node embeddings) and discrete (input graph) features to discrete output features (pooled graph) which continuous output features rely on (new node embeddings). In contrast, these previous approaches focus on mapping purely continuous input features to discrete output features.

Additionally, whereas we opt to sample the perturbed points from the normal distribution, the motivation behind our method does not rely on randomly sampled points and they could easily be chosen in some deterministic way instead. This makes our approach easier to analyze. For instance, if $f$ already corresponds to a hyperplane, for $m > d$ we will always get $f = \hat{f}$ by definition. This holds independent of the choice of perturbed points, as long as they are linearly independent. In methods like the one by Berthet et al. (2020), this only holds in expectation.

### D.2.1 APPLICATION TO GRAPH POOLING

The gradient approximation methods for black box combinatorial solvers that we discussed so far (including our method described in the previous section) are not directly applicable to our hierarchical pooling setting because they assume that the output dimension $d$ of the black box function will be fixed. However, our black box component outputs arbitrarily sized graphs along with their node embeddings. For each pooling layer, we therefore define the black box as its actual discontinuous component followed by all subsequent pool blocks along with the final, global pooling (see Figure 5). Whereas this results in a constant output dimension, it also leads to black boxes containing learnable parameters—a setting which the discussed approximation methods are not applicable for. We therefore always propagate gradients back until we reach a discontinuous component. However, instead of propagating them through this component as described in Section D.1, we compute the gradient with respect to the inputs of the discontinuous component using the black box differentiation method described earlier in this section.

Note that with this method, the number of required forward passes grows in $\Theta(mn_{\text{blocks}})$. Along with the "main" forward pass (for which we calculate gradients), we need $m-1$ additional forward passes from each component up to the final layer. As we do not need to calculate the gradients for these additional forward passes, the number of forward passes does not grow exponentially. Regardless, this still limits us to a relatively low number of samples in practice. In Appendix E we demonstrate that despite yielding relatively stochastic gradients, combined with an optimizer like Adam (Kingma & Ba, 2015), this approach can still lead to convergence. To further stabilize training in our more complex setting, we use a weighted average of the exact gradients with respect to the values (as described in Section D.1) and the gradients obtained by our method, which are stochastic but take different clustering into account.

## E EVALUATION OF HYPERPLANE GRADIENT APROXIMATION

To showcase the smooth gradient estimation generated by our method described in Appendix D.2, we arbitrarily chose the piece-wise linear function $f$ as:

$$f(x_1, x_2) := \begin{cases} 0.2x_1 + 1 & x_1 < 1 \wedge x_2 < -1 \\ 0 & x_1 < 1 \wedge x_2 \geq -1 \\ 4 & x_1 \geq 1 \end{cases} \tag{5}$$

Figure 7 shows that for a higher number of samples we get an increasingly smooth and less stochastic gradient approximation. However, In practice we are limited to a relatively low number of samples in more complex architectures like the one proposed in this thesis. To verify that even noisy gradient estimations allow us to learn, we initialize a point on the red plateau at $(2, -2.5)$ (see Figure 7a) and try to minimize the function using the Adam optimizer (Kingma & Ba, 2015) with learning rate $0.1$ and weight decay $5 \cdot 10^{-4}$. For the gradient estimation, we evaluate at $m = 20$ additional points perturbed by the Normal distribution $\mathcal{N}(0, 0.3)$. Since $f$ does not have a unique minimum, the goal should be arriving at any point with $x_1 < 1, x_2 < -1$ and moving down the slope in the direction of smaller $x_1$ from there. After 1000 steps we end up at $(-76.21, -2.42)$ with $\hat{f}(x_1, x_2)$ monotonically decreasing over all steps, strongly indicating that this is indeed what happens. In particular, despite the theoretical gradient of 0 we are able to leave the plateau and even though the

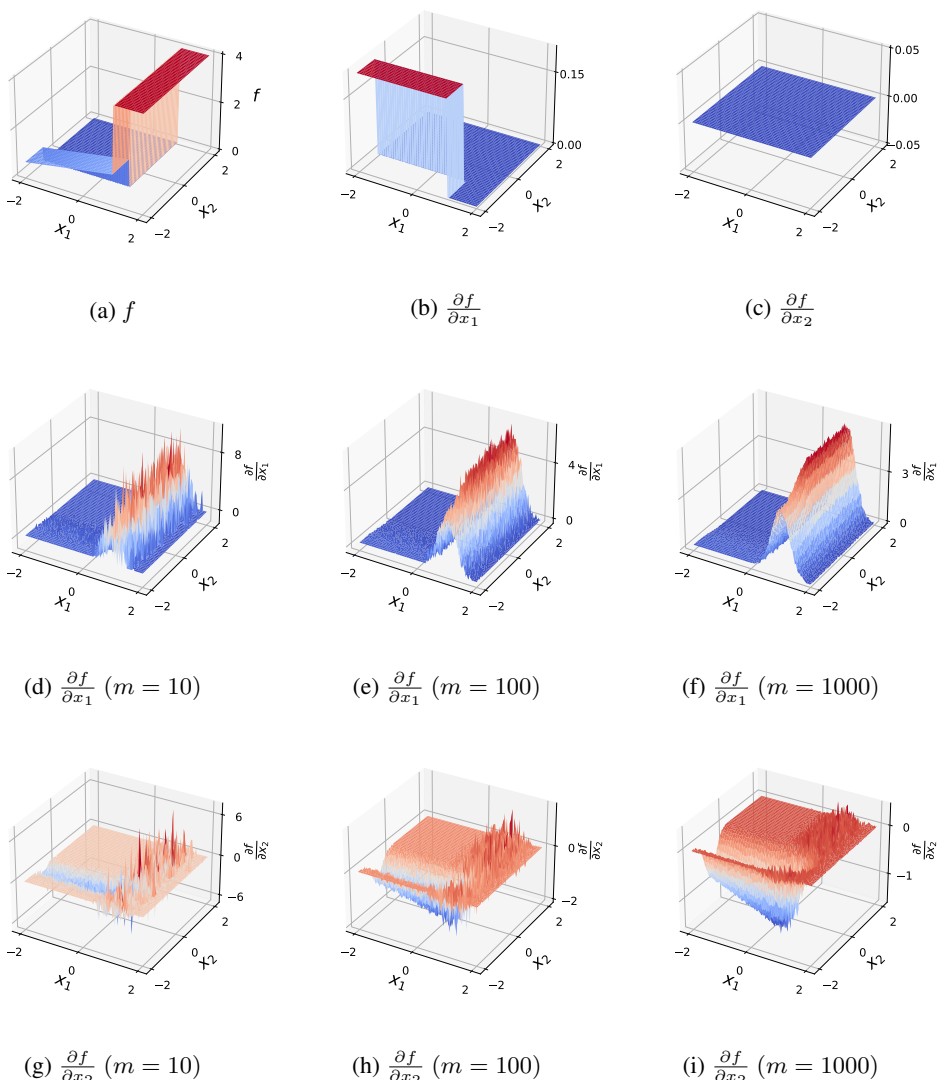

Figure 7: Plot (a) shows the function from Equation 5 with its exact gradients in the directions $x_1$ and $x_2$ in Figures (b) and (c) respectively. Plots (d)–(i) show the corresponding smooth gradient estimations for $m \in \{10, 100, 1000\}$ samples. Perturbations are sampled from $\mathcal{N}(\mathbf{0}, 0.3\mathbf{I}_2)$.

dark blue area $f(x_1, x_2) = 0$ for $x_1 < 1, x_2 \geq -1$ has 0 gradient and a lower value for $x_1 > -5$ we do not end up in this local minimum.

## F  ADDITIONAL RESULTS

### F.1  $L$-HOP NEIGHBORHOOD CONFORMITY

In addition to our definition of conformity, Table 5 also shows the conformity when interpreting examples of a concept as the $L$-hop neighborhood (like proposed in GCExplainer) rather than the connected component of nodes mapped to the same cluster. Crucially, this is not just a different metric definition but rather a question of how we define and therefore interpret concepts. As expected, all conformity scores are significantly lower this way as the subgraphs consist of a fixed neighborhood and will therefore inevitably contain nodes that are not relevant to the concepts, leading to many non-isomorphic subgraphs with the same meaning.

Table 5: **Comparison of concept conformity scores.** Due to the expense of graph-isomorphism tests we only perform one inference pass per trained model (as opposed to 3 for concept completeness). For comparison, we also show the $L$-hop scores (Section F.1). Note that scores close to $100\%$ are expected for DiffPool after the first layer as the pooled graphs look identical for all samples (fully connected with the predefined number of nodes). However, this also yields low completeness scores, meaning that the concepts may be easy to understand (everything is considered the same concept) but not meaningful.

| | **Ours** **Level 1** | $L$-**hop** **Level 1** | **Ours** **Level 2** | $L$-**hop** **Level 2** |
|---|---|---|---|---|
| **Synthetic Hierarchical** | | | | |
| Ours | 99.8%±0.4 | 97.3%±0.2 | 95.0%±2.7 | 89.7%±0.9 |
| Ours (Global Clustering) | **100.0%±0.0** | **97.5%±0.0** | 97.1%±2.2 | 88.8%±3.9 |
| Ours (Hyperplane) | **100.0%±0.0** | 97.1%±0.2 | 97.0%±2.3 | 90.3%±4.0 |
| DiffPool | 0.0%±0.0 | 27.2%±0.0 | **100.0%±0.0** | **100.0%±0.0** |
| **Mutagenicity** | | | | |
| Ours | 83.6%±0.3 | 47.5%±2.4 | 63.5%±4.2 | 23.3%±4.4 |
| Ours (Global Clustering) | **84.2%±1.7** | **47.8%±5.3** | 54.0%±6.3 | 18.2%±5.4 |
| Ours (Hyperplane) | 83.0%±1.2 | 46.7%±3.7 | 60.0%±6.0 | 28.2%±3.9 |
| DiffPool | 42.9%±0.0 | 12.4%±0.0 | **100.0%±0.0** | **100.0%±0.0** |
| **BBBP** | | | | |
| Ours | **84.8%±1.4** | **33.2%±6.0** | 57.5%±2.0 | 7.5%±2.9 |
| Ours (Global Clustering) | 84.4%±2.4 | 30.1%±0.7 | 53.4%±4.0 | 10.5%±4.4 |
| Ours (Hyperplane) | 84.0%±2.0 | 31.3%±2.0 | 59.1%±5.0 | 9.2%±4.7 |
| DiffPool | 0.0%±0.0 | 0.0%±0.0 | **100.0%±0.0** | **100.0%±0.0** |
| **REDDIT-BINARY** | | | | |
| Ours | **96.2%±0.4** | infeasible | 76.0%±11.0 | infeasible |
| DiffPool | 93.0%±2.6 | infeasible | **100.0%±0.0** | infeasible |

The main advantage of taking the $L$-hop neighborhood is that it guarantees, all isomorphic subgraphs mapped to a concept will have exactly the same meaning as they are defined as the *receptive field* of the node. In contrast, in our approach, two concepts could each pool the subgraph "pair of nodes" where in one of them, this stands for the bottom of a house and in the other it stands for the bottom of a triangle. In Section 5.2, we demonstrate that plotting a few example neighborhoods per subgraph is generally sufficient to get a good understanding of meaningful concepts. Note that in practice, GCExplainer ignores node features and tunes the neighborhood size to some $\ell \leq L$ that gives the best concepts which both imply that this property no longer necessarily holds for their reported concepts. Additionally, by ensuring this, the number of subgraphs to analyze is significantly higher which makes understanding each of them infeasible. When it leads to ignoring other concepts, the guaranteed same meaning is no longer beneficial.

## F.2 COMPLETENESS OF HIERARCHICAL CONCEPTS

Table 6: **Test completeness scores after each of the two pooling steps in comparison to other methods.**

|  | **HELP** (Ours) | | DiffPool | |
|---|---|---|---|---|
|  | Level 1 | Level 2 | Level 1 | Level 2 |
| Synthetic | **100.0%±0.0** | **99.5%±1.4** | 27.0%±0.0 | 27.0%±0.0 |
| Mutagenicity | **73.7%±2.7** | **70.8%±2.4** | 53.6%±0.0 | 53.6%±0.0 |
| BBBP | **80.8%±1.4** | **78.3%±2.5** | 77.1%±0.4 | 77.1%±0.4 |
| Synthetic Expressivity | 52.3%±0.0 | **100.0%±0.0** | **53.5%±0.0** | 53.5%±0.0 |

We would generally expect the completeness to be higher in later pooling layers where the concepts incorporate more structural information. However, the trend we observe indicates the opposite. To this end, it is important to note that in our hierarchical setup, these concepts can, in fact, be more meaningful in two ways that are not captured by the completeness measure. Firstly, they aggregate local information that might not be meaningful on its own but could facilitate subsequent layers. For instance, the presence of a pair of carbon atoms might not carry significantly more information towards the final prediction than a single atom and thus not increase the completeness. Yet, the following GNNs would require less layers to detect an aromatic ring from pairs of carbon atoms than they would for the original graph. Secondly, pooling always implies a reduction in the number of nodes. This generally means that the set of concepts is smaller and each of them will therefore need to carry more information in order for the completeness to stay constant.

## F.3 ADDITIONAL VISUALIZATIONS

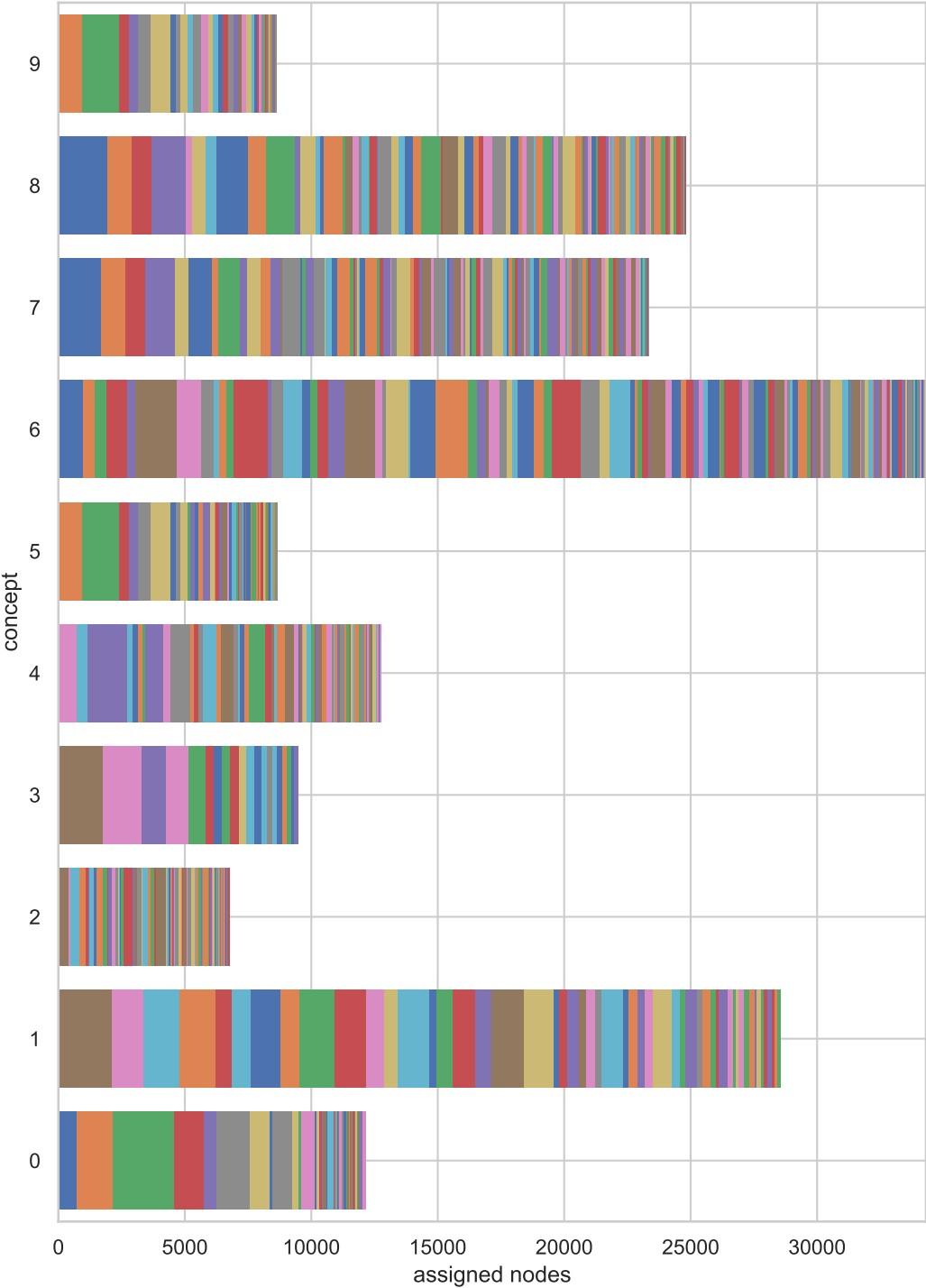

Figure 8: **Subgraphs matched to each concept by GCExplainer in our hierarchical dataset and how often they occur**

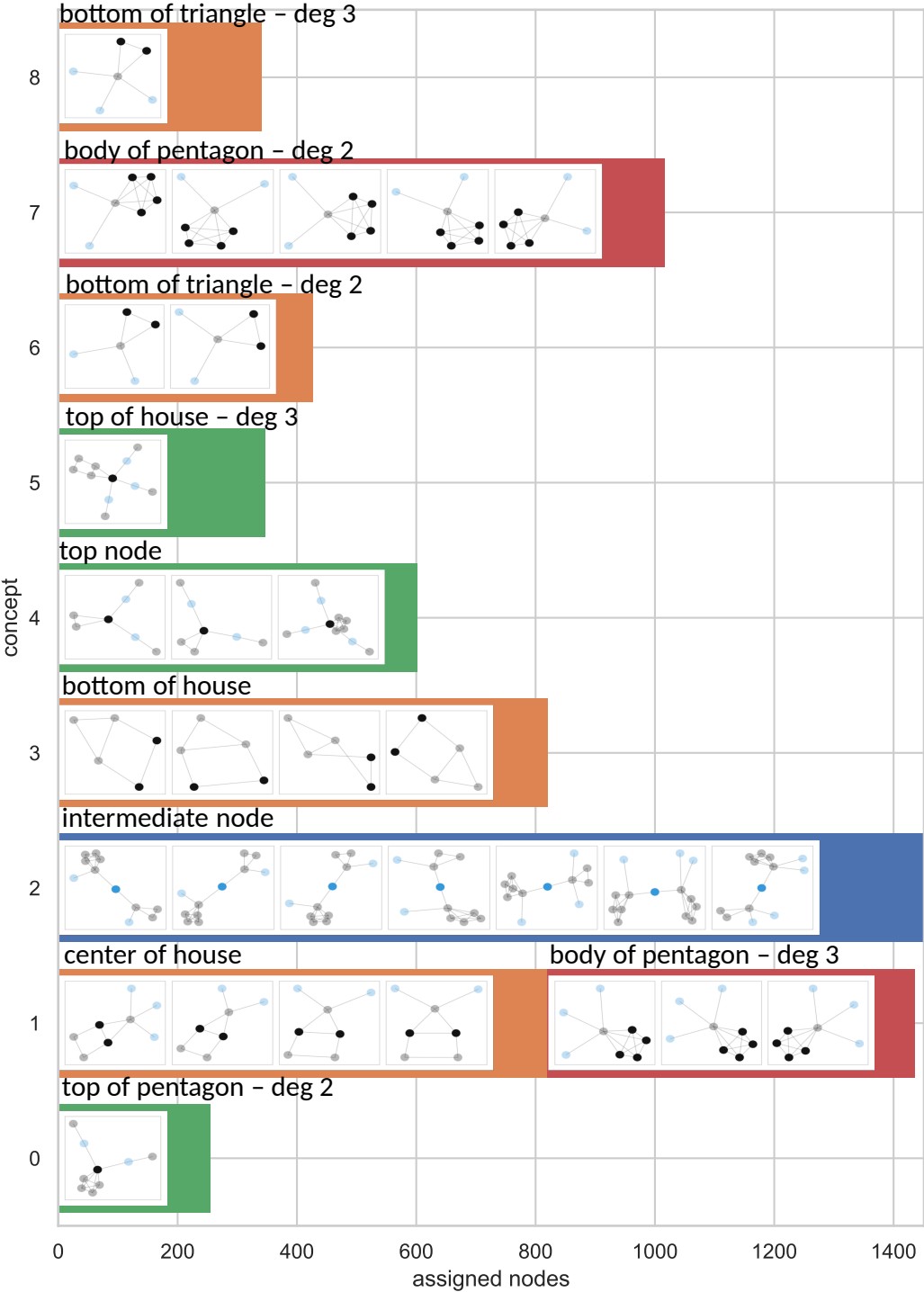

Figure 9: **Subgraphs matched to each concept in the first pooling layer of our hierarchical dataset and how often they occur**

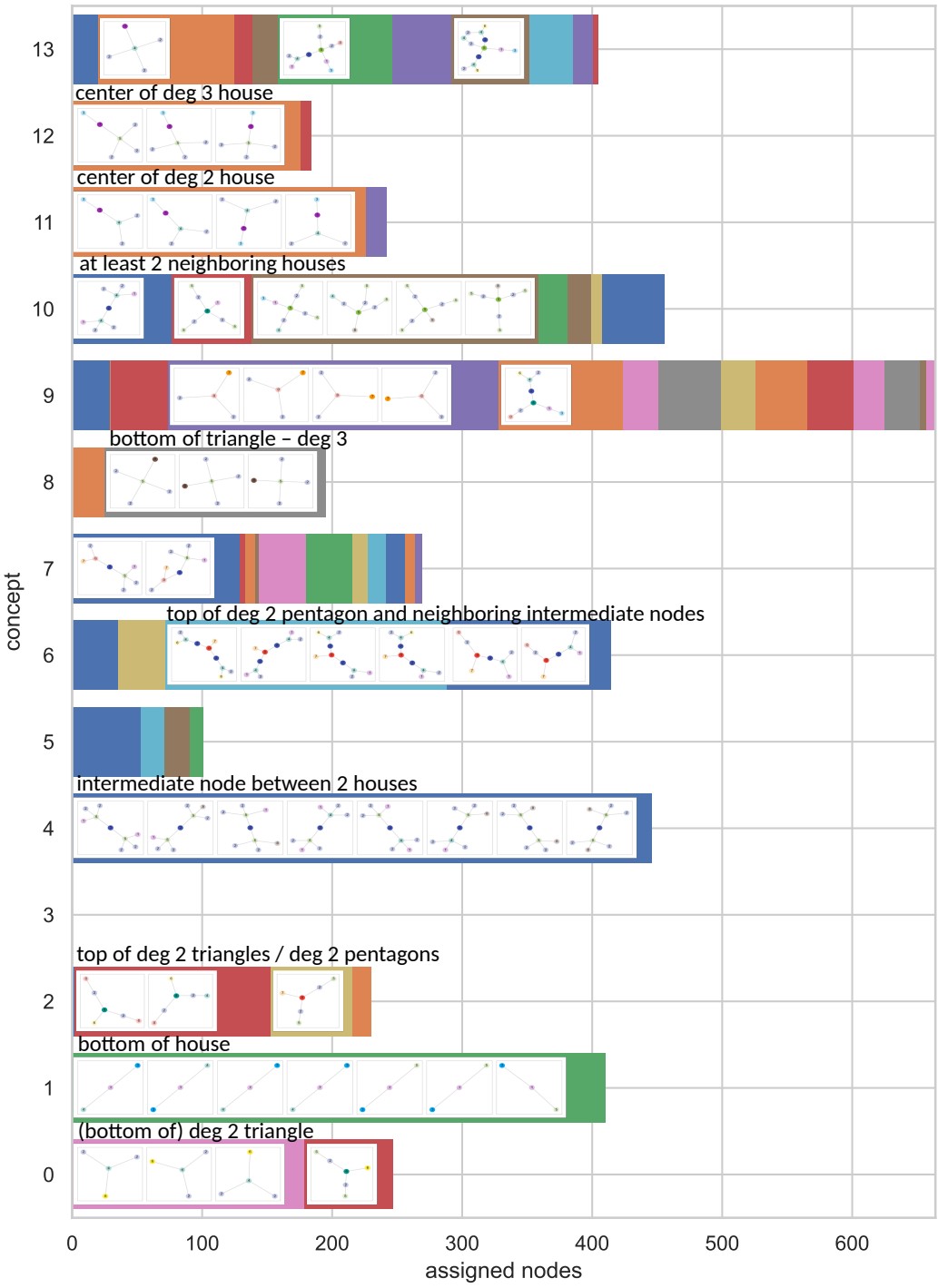

Figure 10: **Subgraphs matched to each concept in the second pooling layer of our hierarchical dataset and how often they occur**

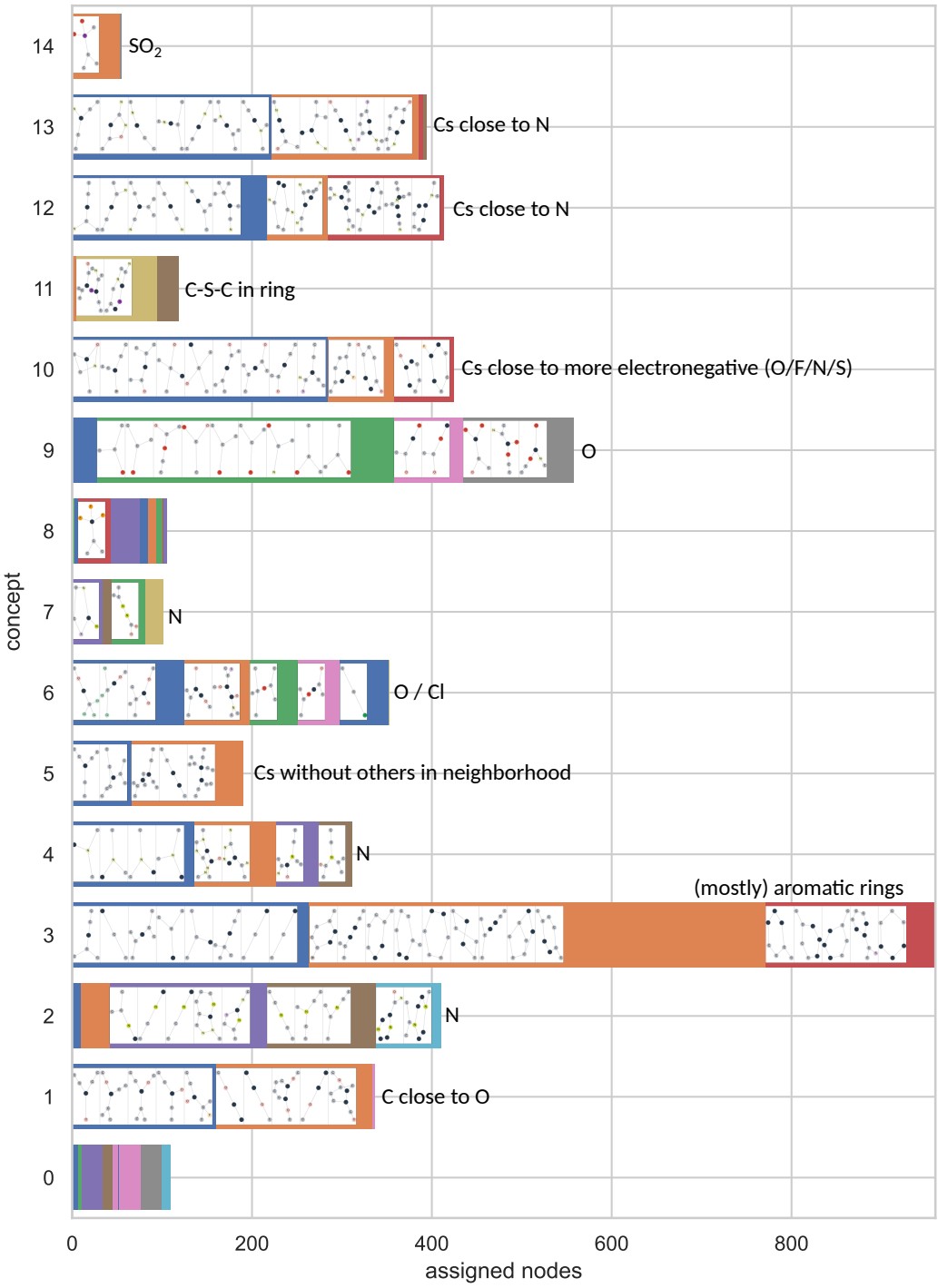

Figure 11: **Subgraphs matched to each concept in the first pooling layer of BBBP and how often they occur**

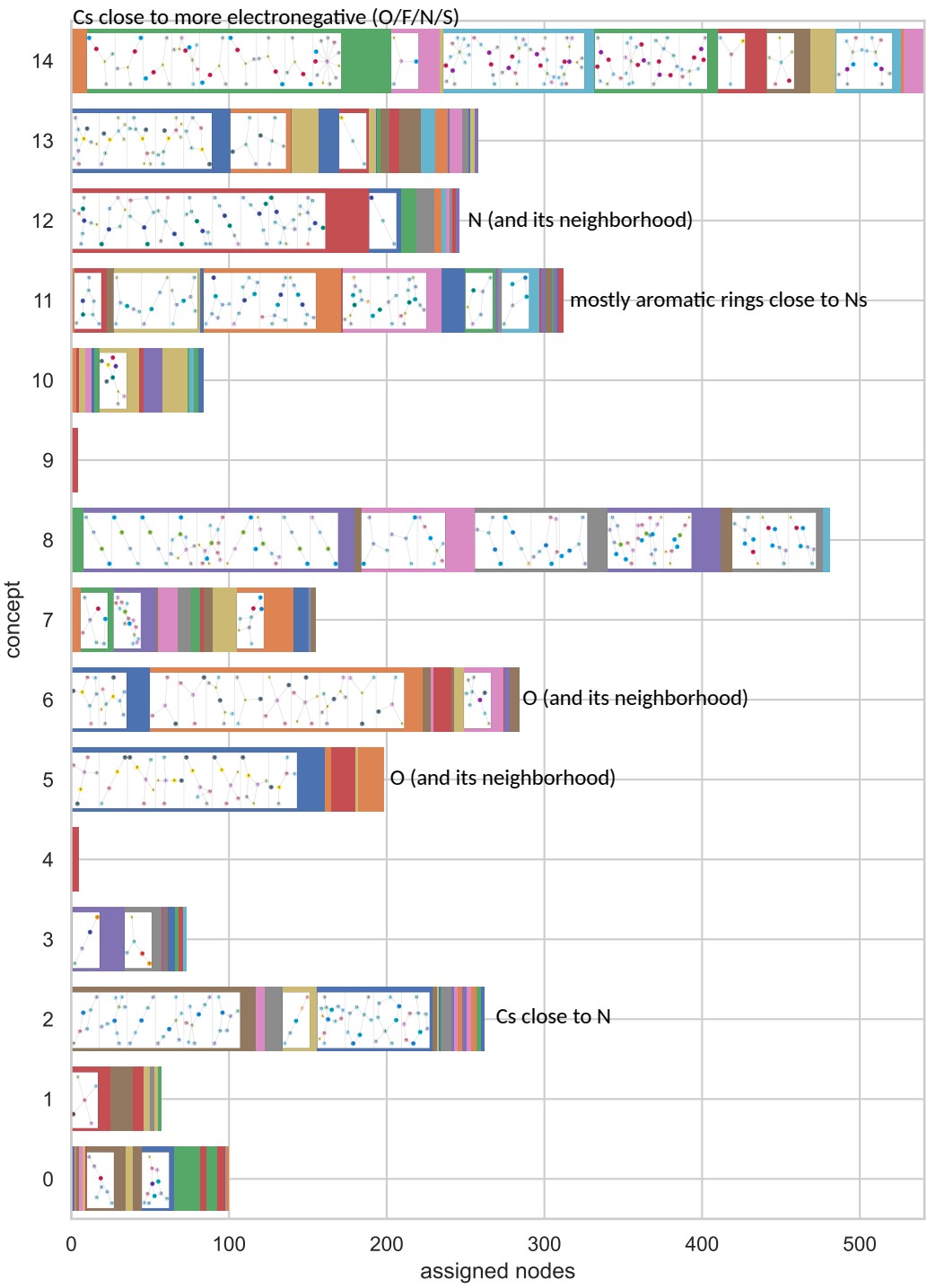

Figure 12: **Subgraphs matched to each concept in the second pooling layer of BBBP and how often they occur**

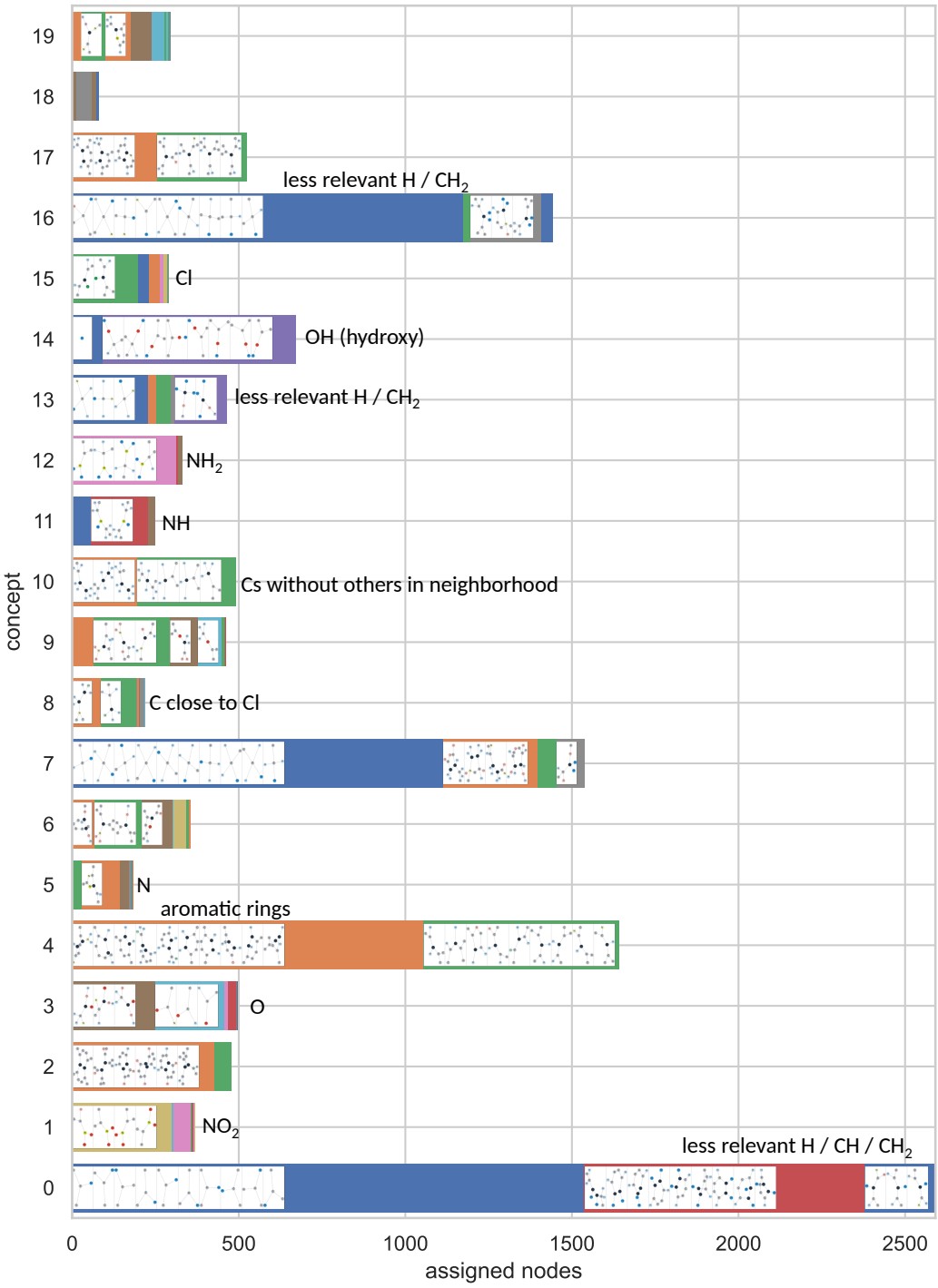

Figure 13: **Subgraphs matched to each concept in the first pooling layer of Mutagenicity and how often they occur**

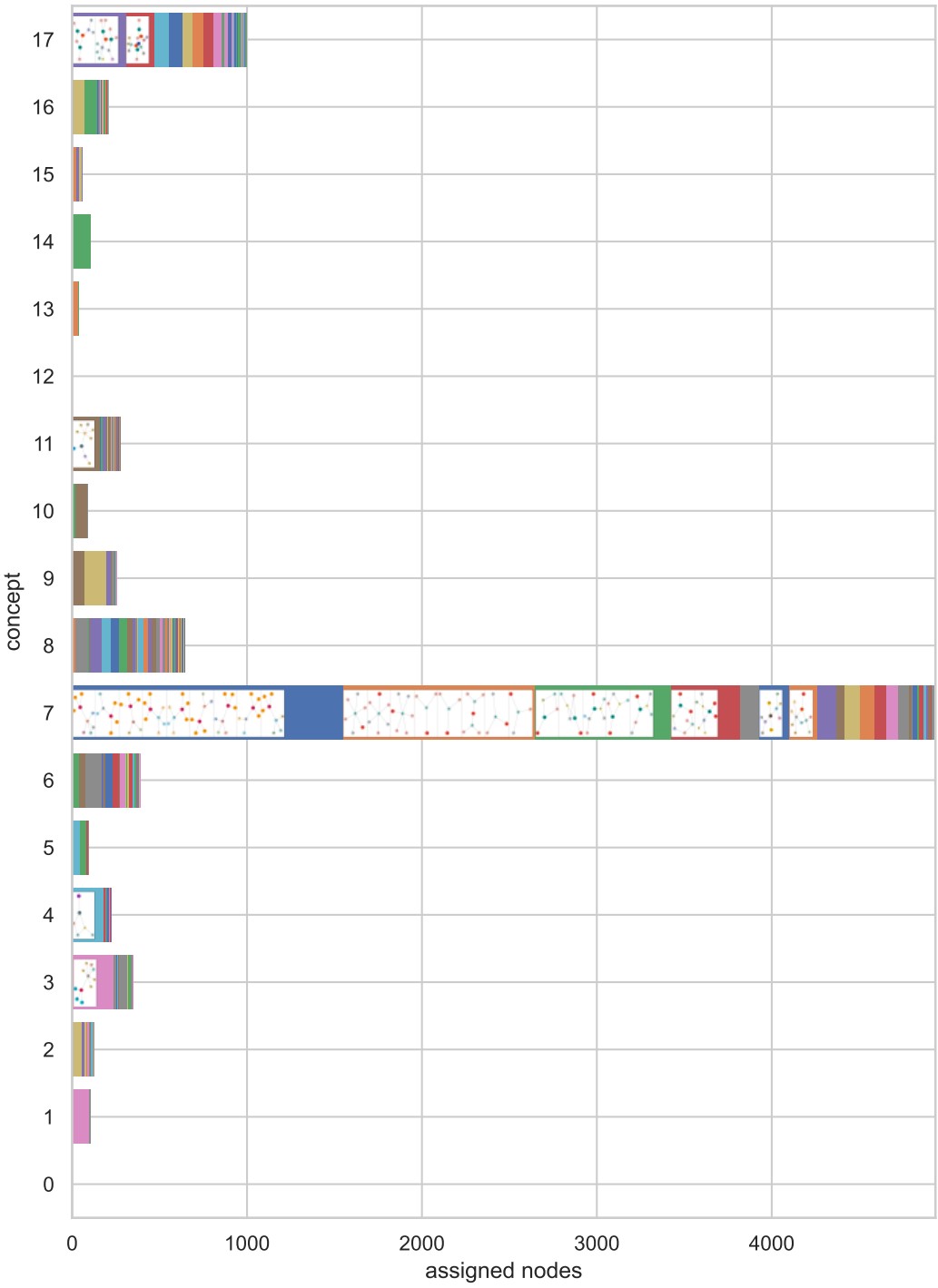

Figure 14: **Subgraphs matched to each concept in the second pooling layer of Mutagenicity and how often they occur**

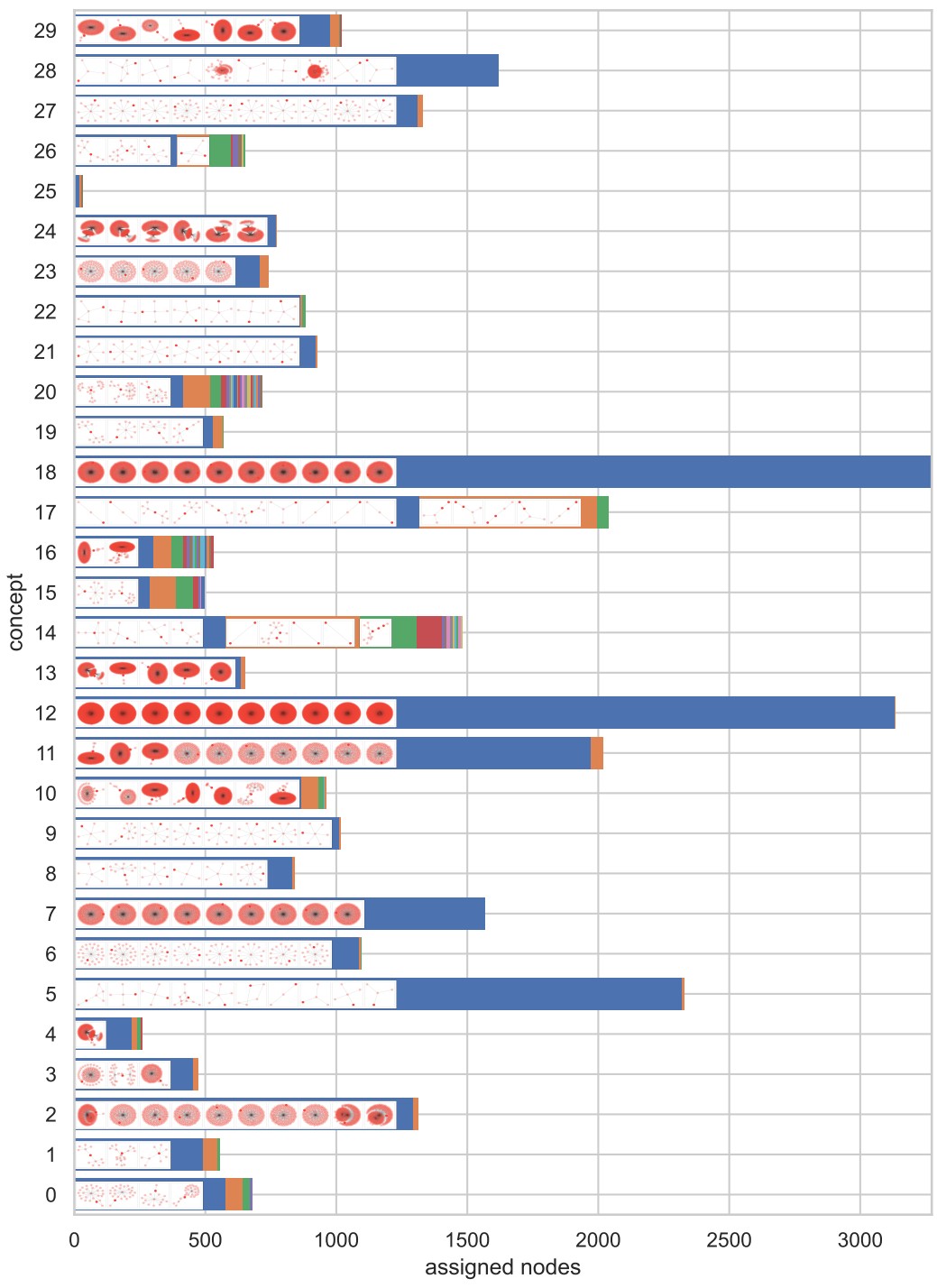

Figure 15: **Subgraphs matched to each concept in the first pooling layer of REDDIT-BINARY and how often they occur.** Generated with 50% of the test data for performance reasons.

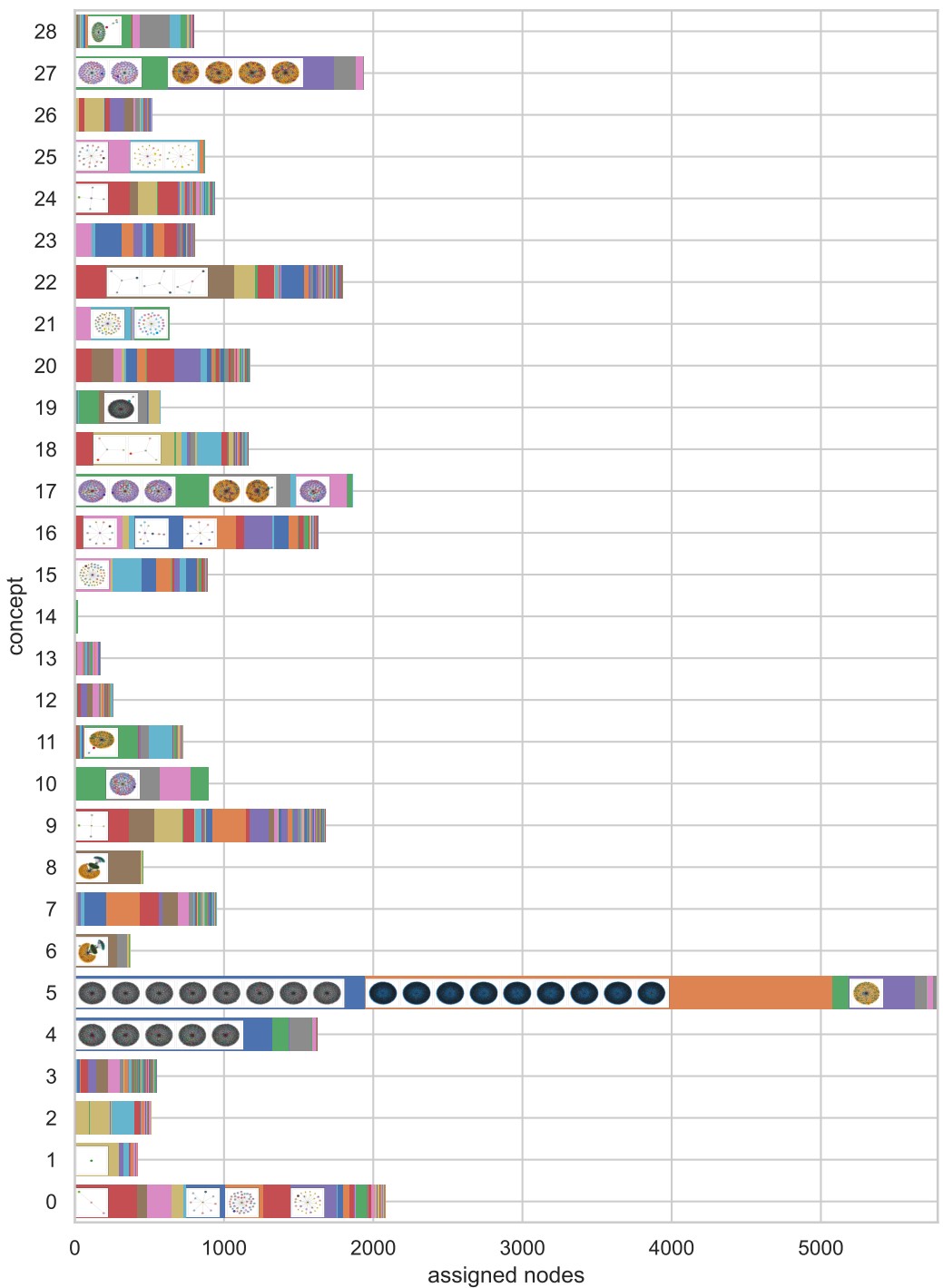

Figure 16: **Subgraphs matched to each concept in the second pooling layer of our REDDIT-BINARY and how often they occur.** Generated with 50% of the test data for performance reasons.

