# OpenReview forum: "Everybody Needs a Little HELP: Explaining Graphs via Hierarchical Concepts"
_ICLR.cc/2024/Conference — Submitted to ICLR 2024_

### Official Review · Reviewer_zgh9 · 2023-10-31

**Soundness:** 2 fair
**Presentation:** 2 fair
**Contribution:** 2 fair
**Rating:** 3
**Confidence:** 4

**Summary:**

The article introduces Hierarchical Explainable Latent Pooling (HELP), an innovative graph pooling method designed to enhance the interpretability of Graph Neural Networks (GNNs). HELP operates by continuously pooling nodes with similar embeddings in the graph, merging these embeddings through average pooling in a hierarchical manner across various levels by k-means. This method allows HELP to identify and elucidate concepts in the input graph pertinent to the model's predictions, with these concepts evolving and increasing in complexity at higher pooling levels. Moreover, this paper designs a new metric Concept Conformity to measure the quality of a concept which is demonstrated better than existing metric in three aspects. Experimental results indicate that HELP matches the performance of leading GNNs while uncovering concepts that are more consistent and in alignment with domain knowledge.

**Strengths:**

1.	designs a new metric Concept Conformity to measure the quality of a concept which is demonstrated better than existing metric in three aspects.
2.	discusses the benefits of using k-means clustering to identify concepts.

**Weaknesses:**

1.	The motivation is not strong enough: Why we need identify concept in hierarchical manner?
2.	The representation need to be improved.
•	The algorithm is very unclear to know how to find concepts.
•	Many typos and undefined symbols such as V_i, b, CONCOMP .
•	Unclear description of the synthetic dataset.
3.	Insufficient baseline.  GLGExplainer (Global Logic-based GNN Explainer) [1] is a post-hoc concept-based explanation method as well.
4.	The experiments do not show the advantage of hierarchical pooling which is the main motivation of this article.
[1] Steve Azzolin, Antonio Longa, Pietro Barbiero, Pietro Lio, and Andrea Passerini. Global explainability of GNNs via logic combination of learned concepts. In The Eleventh International Conference on Learning Representations, 2023.

**Questions:**

1.	For the synthetic dataset, you state “The class label is therefore given by (house, {triangle, house, fully connected pentagon})”, could you explain it in detail?
2.	For the synthetic dataset, what are intermediate nodes?

---

> ### Author Response · Authors · 2023-11-21
> **Official Response to Reviewer zgh9**
>
> Thank you for the feedback on our paper. We have responded to your questions and concerns below.
>
> > Insufficient baseline. GLGExplainer (Global Logic-based GNN Explainer) [1] is a post-hoc concept-based explanation method as well
>
> Indeed, GLGExplainer is one of a variety of existing techniques for concept-based explainability on graphs. In contrast to HELP, it is a post-hoc method rather than an inherently interpretable architecture and it only delivers a set of concepts on the last layer, rather than insights into how concepts compose from the input to the final predictions.
> As HELP is the first method at the intersection of hierarchical graph pooling and interpretability, we chose to cover a selection of both, uninterpretable hierarchical pooling approaches and non-hierarchical methods for explainability.
>
> > Why we need identify concept in hierarchical manner? [...] The experiments do not show the advantage of hierarchical pooling which is the main motivation of this article.
>
> Hierarchies are ubiquitous in graph-structured data (and tasks). Identifying concepts in a more structured manner, allows for better explainability at different levels of granularity: From node-level concepts to concepts on the level of subgraphs which denote different motifs in the data. For instance, as shown in Figures 2a and 2b and explained in Section 5.2, the hierarchical pooling approach allows us to discover how concepts on later layers are composed of concepts from earlier layers. This yields significantly more detailed insights than only a single set of concepts after the final layer (as delivered by previous methods like GCExplainer or CEM (Magister et al. 2022)). Additionally, the hierarchical approach allows us to identify exactly which nodes belong to a concept, rather than just using k-hop neighborhoods. This significantly decreases the number of subgraphs a concept represents (see Figure 2), making it feasible to understand all subgraphs that make up the majority (the percentage is approximated by the concept conformity) of predictions, rather than only giving a few examples of the concept (like, for instance, GCExplainer and CEM).
>
> > The representation need to be improved. • The algorithm is very unclear to know how to find concepts. • Many typos and undefined symbols such as V_i, b, CONCOMP
>
> Thank you for pointing this out. We have improved the notation in the revised version of the paper and added more detailed descriptions of the used symbols. $V_i$ denotes the set of nodes of the $i$th graph in the batch. Please let us know if anything remains unclear. As we were unable to locate the mentioned typos, we would also be grateful if you could clarify which parts you are referring to exactly, so we can address them appropriately.
>
> > For the synthetic dataset, you state “The class label is therefore given by (house, {triangle, house, fully connected pentagon})”, could you explain it in detail? For the synthetic dataset, what are intermediate nodes?
>
> As detailed in Section 4.1 and Appendix C.3.1, the synthetic hierarchical dataset is generated by first sampling a high-level motif (e.g. a house in Figure 4.1). Next, an intermediate node of a special color (node embedding) is inserted on each edge (please refer to Appendix C.3.1 for details on why this is required). Finally, a random low-level motif is sampled for each original node of the high-level motif (without the intermediate nodes). A graph’s class is given by the combination of high-level motif and the set of low-level motifs. In Figure 4.1, the high-level motif is a house and the low-level motifs are two triangles, two fully connected pentagons and a house. The class is therefore given by the tuple (<high-level motif>, <set of low-level motifs>), which is (house, {triangle, house, fully connected pentagon}) in the case of Figure 4.1. In practice, we enumerate all possible classes and aim to predict the class I.D. as a one-hot encoding.
>
> We have added a reference to Appendix C.3.1 in the revised paper.
>
> If you have any further questions or concerns, we would be happy to discuss more!

---

### Official Review · Reviewer_VEwC · 2023-10-31

**Soundness:** 2 fair
**Presentation:** 2 fair
**Contribution:** 2 fair
**Rating:** 3
**Confidence:** 3

**Summary:**

This paper achieved such progress:
1) proposes a new non-spectral interpretable-by-design pooling method called “HELP” to demonstrate how concepts from different GNN layers interplay with each other and compose a new concept in later GNN layers. This method can give explanations to the model prediction in terms of concepts and the analysis of the hierarchical structure of graphs can be performed.
2) proposes a novel metric called “concept conformity” which measures the purity of a given concept to check if the discovered concepts by HELP are meaningful.
3) demonstrate a method of GNN explainability via an interpretable GNN architecture design approach.

**Strengths:**

1. The motivation of this paper is clear, that is to find a GNN explainability method through neural network design and the writing style of this paper is also clear enough for readers to follow.
2. This method offers some inspiration on how to deal with GNN explainability. By searching for high-level explainable concepts, this method can thus identify the relevant subgraphs in the model decision-making process.
3. HELP uses K-Means as part of the algorithm, which is intrinsically more interpretable. Therefore, it can generate a more interpretable explanation of the model prediction compared to other black-box explainers.

**Weaknesses:**

1. The paper lacks context in the introduction of the part 3.2 “gradient flow”, the readers find it hard to understand how this part is related to other parts in this paper. There is no background information about why it is necessary to introduce “gradient flow” in this part and how it relates to other parts of this paper.
2. The explanation and description of how HELP works is insufficient. For instance, In part 3, this paper gives a limited description of how to implement “pooling” in this method (what is the exact way to apply a series of pooling blocks to the input graph, and how they are applied to different GNN layers?). Similarly, the description of Algorithm 1 is limited and readers might be confused about the purpose of each step in this algorithm (e.g., What is CONCOMP and why should we use it in this method?).
3. Though focusing on explainability, the experiment result shows that HELP doesn’t outperform other approaches in model accuracy, so the quality of this method in practice is questioned. The author needs to give more persuasive experiment results to show the feasibility of HELP.
4. The metric concept conformity is not applicable for all methods, e.g., ASAP generates NA for this metric. This means that the applicability of concept conformity requires further exploration.
5. This paper still needs to use other commonly used and standard metrics to measure the quality of generated concepts, instead of solely using two metrics.

**Questions:**

1. Why the discovered concept-based explanations from HELP can give deeper insight compared to previous works? What makes these discovered concepts better compared to previous methods?
2. How to ensure the process of converting graph embedding into a concept explainable enough after multiple layers of GNN? The concepts are generated after many layers of neural networks, so it’s hard to demonstrate that the concepts are still interpretable enough to explain the model prediction.
3. This paper states that “our techniques preserve sparsity” in “paper’s contribution”,  but there is none of any further explanation about this point. How is this statement validated by experiments or theories?
4. The accuracy of HELP always lies in 1 standard deviation from the best approach. Does it necessarily mean that some implementation details in HELP can be further revised to make it perform better?

---

> ### Author Response · Authors · 2023-11-20
> **Response to Reviewer VEwC (1/2)**
>
> Many thanks for your feedback. Below, we address your concerns and provide some clarifications.
>
> > The paper lacks context in the introduction of the part 3.2 “gradient flow”, the readers find it hard to understand how this part is related to other parts in this paper. There is no background information about why it is necessary to introduce “gradient flow” in this part and how it relates to other parts of this paper
>
> By gradient flow, we simply refer to the fact that there exist well-defined gradients from the input to the predictions (which is required for backpropagation). This might not be immediately obvious as we utilize non-differentiable components like clustering. We have amended Section 3.2 to reflect this more clearly.
>
> > The explanation and description of how HELP works is insufficient. For instance, In part 3, this paper gives a limited description of how to implement “pooling” in this method (what is the exact way to apply a series of pooling blocks to the input graph, and how they are applied to different GNN layers?). Similarly, the description of Algorithm 1 is limited and readers might be confused about the purpose of each step in this algorithm (e.g., What is CONCOMP and why should we use it in this method?)
>
> We have clarified our notation in the revised version. Please let us know in case there are any ambiguities left that you would like us to address.
>
> > Though focusing on explainability, the experiment result shows that HELP doesn’t outperform other approaches in model accuracy, so the quality of this method in practice is questioned.
>
> As you noted correctly, the goal of HELP is not to improve model accuracy, but to maintain comparable prediction quality to previous work while delivering insights into why predictions were made.
>
> > The metric concept conformity is not applicable for all methods, e.g., ASAP generates NA for this metric. This means that the applicability of concept conformity requires further exploration.
>
> Concept conformity is a metric designed to evaluate the quality of concepts that a method delivers to explain its predictions. Whereas ASAP is a popular hierarchical pooling approach, it does not deliver any explanations of its predictions. Therefore, there is no sensible way to define concept purity. The reason we use ASAP as a baseline, is to demonstrate that HELP performs comparable while delivering explanations for its predictions.
>
> > This paper still needs to use other commonly used and standard metrics to measure the quality of generated concepts, instead of solely using two metrics.
>
> Unfortunately, there are few commonly adapted metrics for this at the moment. The most popular one is concept completeness (which we report). Additionally, we report concept conformity, which measures the same objective as concept purity (proposed in GCExplainer), in a different way as justified in Section 4.2. In Table 5, we also give conformity scores using the k-hop neighborhood which makes them more similar to concept purity. Please let us know if there are any other metrics you had in mind.
>
> > Why the discovered concept-based explanations from HELP can give deeper insight compared to previous works? What makes these discovered concepts better compared to previous methods?
>
> As we will detail in the next question, an important advantage of HELP is that it discovers a hierarchy of concepts, giving detailed insights over all GNN layers, rather than merely the result of the last layer (like, for example, GCExplainer). Additionally, comparing Figures 2a and 2b to Figure 2c clarifies the practical impact of the higher concept conformity. Concepts discovered by HELP represent one of a small set of subgraphs, making it easy to understand what a concept stands for. In contrast, concepts discovered by GCExplainer represent a significantly larger set of different subgraphs, making it almost impossible for a human to look at all of them and identify a common theme or meaning of the concept.

---

> ### Author Response · Authors · 2023-11-20
> **Response to Reviewer VEwC (2/2)**
>
> > How to ensure the process of converting graph embedding into a concept explainable enough after multiple layers of GNN? The concepts are generated after many layers of neural networks, so it’s hard to demonstrate that the concepts are still interpretable enough to explain the model prediction.
>
> Indeed, you are  pointing out an important advantage of HELP. As you mention, existing, concept-based methods only generate a set of concepts after the last layer. In contrast, HELP generates a set of concepts after each pool block. As nodes generated from a similar concept/cluster, have a similar embedding¹, we can view concepts from a given pool block as a set of (sub)graphs over concepts from the previous block (see Section 5.2). Note that in our experiments, there are only 2 GNN layers per pool block (see Table 3).
>
> > This paper states that “our techniques preserve sparsity” in “paper’s contribution”, but there is none of any further explanation about this point. How is this statement validated by experiments or theories?
>
> By design, HELP only merges connected components and never creates any edges. It therefore preserves sparsity in the sense that a sparse input graph will not generally be fully connected after pooling, as it would be, for instance in DiffPool.
>
> > The accuracy of HELP always lies in 1 standard deviation from the best approach. Does it necessarily mean that some implementation details in HELP can be further revised to make it perform better?
>
> Indeed, it is possible that the performance of HELP could be improved by a thorough hyperparameter search, which is beyond the capacity of our resources. This seems particularly likely for the hyperplane gradient approximation as discussed in the Limitations in Section 5.3.
>
> We would be happy to discuss more if you have any further questions or concerns!
>
> ---------------------
> ¹the average over multiple nodes in the same cluster is expected to be close to the centroid

---

> > ### Comment · Reviewer_VEwC · 2023-11-23
> > **Response**
> >
> > Thank you for your rebuttal. I have reviewed your response along with other reviews and rebuttals. I will keep my current score.

---

### Official Review · Reviewer_dvgz · 2023-10-31

**Soundness:** 3 good
**Presentation:** 2 fair
**Contribution:** 2 fair
**Rating:** 5
**Confidence:** 3

**Summary:**

This paper introduces a hierarchical pooling procedure. At each step, the model processes multiple GNN layers at each step, performs clustering on representation, and merges connected components within the same cluster. By analyzing the relevant node mergers, we can gain insights into the model's decision-making process. In addition, this paper proposes a novel metric concept conformity to measure the noise level in the discovered concepts.

**Strengths:**

1) The proposed method takes into account the interactions between GNN layers, capturing the model's reasoning process from a hierarchical perspective, which refines the interpretability. In Section 3.1, the mentioned global clustering and merging clusters can enhance the effectiveness of clustering.

2) The paper is well written and easy to understand.

3) The paper studies an interesting problem that helps explain graphs.

**Weaknesses:**

1) In the experimental section, a synthetic hierarchical dataset is used instead of the commonly used BA-Shapes and BA-Community datasets. Since conventional datasets do not exhibit a hierarchical structure, does this mean that the proposed method cannot be applied to these datasets and real-world datasets, and therefore has limitations? In Table 1, for the real-world dataset, HELP does not perform very well.

2) The proposed synthetic hierarchical dataset is worth discussing. The accuracy of the model may be so high that the prediction of the dataset may be easier.

3) Intuitively, as the number of layers in the model increases, the model will capture fine-grained information. However, the method proposed in this paper pools the input graph to a coarser representation, so does it ignore the fine-grained features of the nodes at high-level layer?

4) For the metric Concept Conformity, i don't understand the formula, from the interpretation of the formula conf(c) should always be 1. In addition, will there be a case where two noise clusters are larger than the threshold t after merging, and then what should be done with these noise clusters?

**Questions:**

Please see the questions given in the weakness part.

---

> ### Author Response · Authors · 2023-11-20
> **Official Response to Reviewer dvgz**
>
> Thank you for your review and your supportive comments about our paper! We would like to address your questions and concerns below.
> > Since conventional datasets do not exhibit a hierarchical structure, does this mean that the proposed method cannot be applied to these datasets and real-world datasets, and therefore has limitations?
>
> We would like to clarify that most real-world graphs are generally expected to exhibit hierarchical structure. For instance, molecular graphs like in BBBP or Mutagenicity can be seen as graphs over functional groups rather than atoms, as explained in Figure 1a. This is the underlying assumption of existing hierarchical pooling methods like ASAP and DiffPool and is supported by the concepts discovered by our method. The reason why we construct our hierarchical dataset in the given way is that in real-world datasets, the ground-truth hierarchies are unknown. This makes our synthetic dataset a useful sanity check to verify that HELP discovers the expected structures.
>
> > The proposed synthetic hierarchical dataset is worth discussing. The accuracy of the model may be so high that the prediction of the dataset may be easier.
>
> We agree that our hierarchical synthetic dataset may be easier to learn than some of the real-world datasets. As mentioned above, its primary purpose is to verify the generated explanations in a scenario where the ground-truth structure is known, making it easy to recognize valid explanations.
>
> > Intuitively, as the number of layers in the model increases, the model will capture fine-grained information. However, the method proposed in this paper pools the input graph to a coarser representation, so does it ignore the fine-grained features of the nodes at high-level layer?
>
> Ideally, the parts of the fine-grained structure which are relevant for the prediction would get incorporated into the node embedding of the pooled node, making it available for the subsequent layers.
>
> > For the metric Concept Conformity, i don't understand the formula, from the interpretation of the formula conf(c) should always be 1.
>
> Thank you for pointing this out. Indeed, we seem to have lost the crucial indicator function in the editing process which would make the conformity always 1 as you recognized correctly. We corrected the formula in the updated version.
>
> > Will there be a case where two noise clusters are larger than the threshold t after merging, and then what should be done with these noise clusters?
>
> We assume you are referring to merging clusters as described in Section 3. First, we want to emphasize that the threshold $t$ is applied to the percentage of components mapped to a given cluster which were isomorphic, not to the size of the cluster compared to others. That said, if the percentage of occurrences of a particular component is below the threshold in two clusters ($\frac ab < t$, $\frac cd < t$), then it will also be below the threshold after merging those clusters ($\frac{a+c}{b+d}<t$).
>
> We are open and happy to discuss further if you have any more questions or concerns!

---

### Official Review · Reviewer_RY8e · 2023-11-04

**Soundness:** 2 fair
**Presentation:** 2 fair
**Contribution:** 2 fair
**Rating:** 6
**Confidence:** 3

**Summary:**

The paper introduces HELP (Hierarchical Explainable Latent Pooling), a new graph pooling method that enhances the interpretability of Graph Neural Networks (GNNs) by elucidating how concepts across different GNN layers combine to form complex representations. HELP is the first non-spectral, end-to-end learnable hierarchical graph pooling method that can handle a variable number of connected components and demonstrates competitive accuracy with standard GNNs. The method's efficacy is quantitatively validated using novel metrics like concept completeness and conformity and qualitatively through expert-aligned explanations in domains such as chemistry and social networks, marking a significant advancement in making GNNs more interpretable.

**Strengths:**

1. Addressing the Challenge of Explainability in Graph Neural Networks:
The paper tackles a pressing and highly relevant issue in the field of graph learning – the need for explainability. Explainability is crucial in deploying GNNs for real-world applications where understanding the model's decision-making process is as important as the accuracy of its predictions.

2. Innovative Approach to Learning Hierarchical Structures:
The introduction of a hierarchical structure to improve explainability is a novel and compelling approach. Hierarchical interpretations of data are more aligned with human cognitive processes, making them a natural fit for explainability purposes.

**Weaknesses:**

1. Inadequate Methodological Detail:
The paper falls short in providing essential details in the method section, notably omitting some crucial symbols and function definitions (See Question 1). This lack of clarity impedes the reader's ability to fully comprehend the proposed interpretative framework for graph neural networks.

2. Insufficient Coverage of Related Works:
Although the paper mentions the DiffPool method, it neglects to discuss other works in the domain of learnable pooling in GNNs. Specifically, the paper [1] also proposed a learnable clustering approach which is highly relevant to the context of the presented work.

[1] Brain Network Transformer, NeurIPS 2022

3. Over-simplification of Synthetic Datasets:
The nearly perfect classification accuracy on synthetic datasets raises concerns about the complexity and applicability of the test environment. Such high performance suggests that the synthetic dataset may be too simplistic to effectively challenge and evaluate the proposed model.

4. Lack of Explicit Demonstration of Explainability Improvement:
The paper posits that high concept conformity leads to improved explainability but fails to demonstrate this relationship concretely. To substantiate such claims, the authors should provide empirical evidence or case studies that illustrate how explainability is enhanced as a direct result of increased concept conformity (See Question 3).

**Questions:**

1. In the Algorithm Section, please provide detailed definitions of used symbols and functions, for example, n_{blocks}, C, CONCOMP().
2. Please enlarge the font size in Figure 2, and can you explain the X and Y axes in detail?
3. Can you provide a case study to show which benefit the high Concept conformity can bring?

---

> ### Author Response · Authors · 2023-11-20
> **Official Response to Reviewer RY8e**
>
> Thank you for your feedback and time spent reviewing our work. Below, we address each of your comments and questions.
>
> > The paper falls short in providing essential details in the method section, notably omitting some crucial symbols and function definitions. [...]. In the Algorithm Section, please provide detailed definitions of used symbols and functions, for example, n_{blocks}, C, CONCOMP()
>
> Thank you for pointing this out. We have added more detailed descriptions of the methods and variables in use. We hope these modifications address your concerns.
>
> > Insufficient Coverage of Related Works: [...] Specifically, the paper [1] also proposed a learnable clustering approach which is highly relevant to the context of the presented work.
>
> Thank you for making us aware of this work. We have added it to the related work section.
>
> > Please enlarge the font size in Figure 2, and can you explain the X and Y axes in detail?
>
> We will aim to improve readability of the figure in the final version. The y axis denotes a unique id of the cluster/concept. It therefore displays the numbers $0$ to $k-1$ (where $k$ is the number of clusters in the k-means clustering). For brevity, we only show a selection of these concepts in Figure 2 and refer to Appendix F.3 for plots that contain all concepts. The x axis denotes the number of nodes in the test set which were mapped to this particular concept. Nodes of the same color were mapped to the same component. For instance, all nodes that were pooled together in the same functional group would be mapped to the same color.
>
> > The paper posits that high concept conformity leads to improved explainability but fails to demonstrate this relationship concretely. To substantiate such claims, the authors should provide empirical evidence or case studies that illustrate how explainability is enhanced as a direct result of increased concept conformity [...]. Can you provide a case study to show which benefit the high Concept conformity can bring?
>
> Intuitively, concept conformity measures what percentage of the colored segments in a plot similar¹ to Fig. 2 are bigger than some threshold. With a threshold of $t:=10%$ and the example of functional groups, a conformity of 100% implies that at most 10 (usually less) functional groups can be mapped to this concept, which makes it easy to understand by looking at the small number of functional groups it represents. 90% conformity would imply that 90% of the connected components mapped to this concept are functional groups that made up at least 10% of the total number of components. Therefore, 90% of the occurrences can be explained by at most 9 subgraphs. The rest could be a larger number of more rare subgraphs which might make it harder to individually consider all of them. A purity of 0% would imply that the concept can stand for many different subgraphs, making it hard to understand.
>
> We are open to discuss further if you have any more questions or concerns!
>
> ---------------------
> ¹The only difference is that the x axis would be the number of components rather than nodes, i.e. the size of each bar would be divided by the size of the component (e.g. 3 in a functional group with 3 atoms like NO2)

---

### Author Response · Authors · 2023-11-21
**Summarized Response to all Reviewers**

We are grateful to all the reviewers for their comments, suggestions, and feedback.

In our response, we addressed most of the specific concerns to each reviewer, separately. However, we also identified two, more general, concerns   raised by several reviewers that we would like to address centrally. First, we acknowledge that some notations in the paper and in the algorithm were, in parts, ambiguous. We thank the reviewers for bringing this to our attention. We  clarified and highlighted the important parts in the revised manuscript.

Second, we would like to again emphasize the main advantages of HELP over existing concept-based explainability methods. HELP, in contrast to other methods, is able to discover concepts at multiple GNN layers, explaining how the important components of the input hierarchically combine within the final predictions over the course of the whole model. This offers significant benefits, which complement the good quality of the discovered concepts (e.g. in terms of conformity). Therefore, HELP goes beyond the capabilities of existing techniques like GCExplainer, CEM, or GLGExplainer, which only deliver a set of concepts for the state after the final GNN layer. These combined performance-explainability benefits extend towards many practical applications that underline large graph data which includes clusters (or communities) of related samples, such as the ones found in molecular biology, healthcare, traffic data etc.

We hope to continue the discussion further if you have any more concerns.

---

### Meta-Review · Area_Chair_ABGg · 2023-12-10

**Metareview:**

The paper proposes HELP, an algorithm for pooling a GNN in order for inherently interpretable predictions. I believe that both the general motivation and idea of the paper are interesting and timely, as GNNs are increasingly being applied in many domains. However, I believe that the paper need of further improvements and another round of reviews before being ready for publication. I have carefully read the paper, reviews, and discussions with the reviewers, and while I acknowledge the effort made by the authors to clarify their contributions, I believe that most of the issues are consequence of the motivation and presentation of their approach.

First, I believe that the authors should better position their approach in the context of existing literature, as well as provide some qualitative results (e.g., with real examples similar to the one in Figure 1a)) to better exemplify the piratical utility of their approach. The authors claim that the main advantage of their approach lie in the interpretability of their results, however, it remains unclear to me that this is the case and also how the proposed method can be used. Figure 2 and Appendix F3 seems to attempt to show an example, however, it is unclear where the concept labels assigned to each cluster come from and how the authors make sure that they are faithful to the predictions and also interpretable by humans.

Second, I miss theoretical results (or at least discussion) on the optimality (and even convergence) of the proposed pooling approach. It is well known that K-means is pruned to local optima, also it seems that the gradient for updating the weights may easily get stuck at local optima. So I wonder, whether the authors have any solution for finding a good solution for their algorithm beyond hyper parameter tuning and validation performance.

In summary, I believe that the paper adopts a promising direction to obtain inherently interpretable GNNs, but further clarification and polishing is necessary to clearly state (as well as enable a proper assessment of) the contribution and practical implications of the proposed algorithm.

**Justification For Why Not Higher Score:**

Reasonable idea with lacking details on motivation and practical implications that result in a paper, which in my opinion needs further polishing before being ready for publication, and thus is below the ICLR acceptance bar.

**Justification For Why Not Lower Score:**

N/A

---

### Decision · Program_Chairs · 2024-01-16

Reject